



# 1 The export of African mineral dust across the Atlantic
# 2 and its impact over the Amazon Basin

Xurong Wang[1,2,a,+], Qiaoqiao Wang[1,2,+], Maria Prass[3], Christopher Pöhlker[3], Daniel
Moran-Zuloaga[3], Paulo Artaxo[4] Jianwei Gu[5], Ning Yang[1,2], Xiajie Yang[1,2],
Jiangchuan Tao[1,2], Juan Hong[1,2], Nan Ma[1,2], Yafang Cheng[3], Hang Su[3], Meinrat O.
Andreae[3,6]
[1.] Institute for Environmental and Climate Research, Jinan University, Guangzhou, 511443, China
[2.] Guangdong-Hongkong-Macau Joint Laboratory of Collaborative Innovation for Environmental
Quality, Guangzhou, 511443, China
[3.] Multiphase Chemistry Department, Max Planck Institute for Chemistry, Mainz, 55128, Germany
[4.] Institute of Physics, University of São Paulo, São Paulo, 05508-900, Brazil
[5.] Institute of Environmental Health and Pollution Control, School of Environmental Science and
Engineering, Guangdong University of Technology, Guangzhou, 510006, China
[6.] Scripps Institution of Oceanography, University of California, San Diego, CA 92093-0230, USA
[a.] now at: Atmospheric Chemistry Department, Max Planck Institute for Chemistry, Mainz, 55128,
Germany
[+.] These authors contribute equally to this article
Correspondence to, Qiaoqiao Wang (qwang@jnu.edu.cn)
**Abstract**
The Amazon Basin is frequently influenced by the trans-Atlantic transport of African
dust plumes during its wet season (January – April), which not only interrupts the
near-pristine atmospheric condition in that season, but also provides nutrient inputs
into the Amazon rainforest associated with dust deposition. The factors controlling the
long-range transport (LRT) of African dust towards the Amazon Basin and
consequently the overall impact of African dust over the Amazon Basin are not yet
well understood. In this study, we use the chemical transport model GEOS-Chem to
investigate the impact of the export of African mineral dust upon the Amazon Basin
during the period of 2013 – 2017, constrained by multiple datasets obtained from
AERONET, MODIS, as well as Cayenne site and the Amazon Tall Tower Observatory
(ATTO) site in the Amazon Basin. With optimized particle mass size distribution
(PMSD), the model well captures observed AOD regarding both the mean value as
well as the decline rate of the logarithm of AOD over the Atlantic Ocean along the
transport path (AOaTP), implying the consistence with observed export efficiency of



35 African dust along the trans-Atlantic transport. With an annual emission of $0.73 \pm$

36 $0.12$ Pg a$^{-1}$, African dust entering the Amazon Basin has surface concentrations of $5.7$

37 $\pm 1.3$ µg m$^{-3}$ (up to 15 µg m$^{-3}$ in the northeast corner) during the wet season,

38 accounting for $47\% \pm 5.0\%$ (up to 70%) of mass concentrations of total aerosols. The

39 frequency of dust events in the Amazon Basin (defined as when surface dust

40 concentrations reach the threshold of 9 µg m$^{-3}$ on daily basis) in the wet season is

41 around 18% averaged over the basin, with maxima over 60% at the northeast coast.

42 During the dust events, AOD over most of the Amazon Basin is dominated by dust.

43 Observed dust peaks over the Amazon Basin are generally associated with relatively

44 higher African dust emissions (including Sahara and Sahel) and longer lifetime of dust

45 along the trans-Atlantic transport, namely higher export efficiency of African dust

46 across the Atlantic Ocean. Associated with dust deposition, we further estimate annual

47 inputs of $52 \pm 8.7$, $0.97 \pm 0.16$ and $21 \pm 3.6$ mg m$^{-2}$ a$^{-1}$ for iron, phosphorus and

48 magnesium deposited into the Amazon rainforest, respectively, which may well

49 compensate the hydrologic losses of nutrients in the forest ecosystem.

50

51 **1 Introduction**

52 The desert over North Africa, being the world's largest dust source, contributes to

53 over 50% of global dust emission (Kok et al., 2021; Wang et al., 2016). Dust particles

54 are uplifted by strong surface winds, and then typically transported downwards for a

55 long distance, reaching Atlantic, Caribbean, America and Europe (Prospero et al.,

56 1981; Ben-Ami et al., 2012; Yu et al., 2019; Swap et al., 1992; Prospero et al., 2014;

57 Wang et al., 2020). The emission varies on daily to seasonal and even decadal time

58 scales, largely affected by local wind speed, land surface cover, soil moisture, etc

59 (Ridley et al., 2014; Mahowald et al., 2006). Once present in the atmosphere, mineral

60 dust can degrade air quality downwind as well as modify the atmospheric radiative

61 balance via directly scattering and absorbing solar radiation (Ryder et al., 2013b), and

62 altering cloud properties by acting as cloud condensation nuclei or ice nuclei (Chen et

63 al., 1998; Demott et al., 2003; Mahowald and Kiehl, 2003; Dusek et al., 2006).





Additionally, mineral dust contains iron, phosphorous and other nutrients, and could
affect ocean biogeochemistry and fertilize tropical forest upon downwind deposition
(Niedermeier et al., 2014; Rizzolo et al., 2017).
There is an increased concern about the impact of African dust exerted over the
Amazon basin, which being the world's largest rainforest, represents a valuable but
also vulnerable ecosystem, and is sensitive to any disturbance from climate changes
associated with human activities in the future (Andreae et al., 2015; Pöhlker et al.,
2019). During the wet season (January – April) Amazonian aerosols are generally
dominated by local biogenic aerosol, with remarkably low $PM_{10}$ mass concentrations
of a few $\mu g\ m^{-3}$ (Andreae et al., 2015; Martin et al., 2010a; Prass et al., 2021). The
near-pristine condition, however, is frequently interrupted by the transatlantic
transport of African dust toward the Amazon Basin (Andreae et al., 2015; Martin et
al., 2010b; Martin et al., 2010a; Talbot et al., 1990). The dusty episodes could
drastically increase the aerosol optical depth (AOD, by a factor of 4), mass
concentrations of coarse aerosol (with diameter > 1 $\mu m$) (up to 100 $\mu g\ m^{-3}$), as well as
crustal elements over the Amazon Basin (Ben-Ami et al., 2010; Pöhlker et al., 2019;
Moran-Zuloaga et al., 2018; Baars et al., 2011; Formenti et al., 2001). Therefore, there
is great interest in understanding factors controlling the export of African dust towards
the Amazon Basin and the impact they might have on the environment, ecosystem,
and climate.
Over the past decades, field measurements combined with satellite observation and
forward/back trajectory analysis have been conducted to explore the long-range
transport (LRT) of African dust toward the Amazon Basin (e.g. Ben-Ami et al., 2010;
Pöhlker et al., 2018; Prospero et al., 2020). The transatlantic transport of African dust
plumes is closely related to annual north-south oscillation of the intertropical
convergence zone (ITCZ) (Moran-Zuloaga et al., 2018; Ben-Ami et al., 2012),
favoring the path towards the Amazon Basin in the late boreal winter and spring
(December-April) as the ITCZ moves southward. In addition to the annual oscillation
of ITCZ, the export efficiency of African dust towards the Amazon Basin also highly



93 depends on the lifetime of mineral dust, which is largely affected by the

94 meteorological condition (e.g. precipitation). Dust particles are subject to wet removal

95 when they are within or underneath precipitating clouds. For instance, Yu et al. (2020)

96 argued that El Djouf contributes more dust to the Amazon Basin than the Bodélé

97 depression as the transport paths of dust released from El Djouf are less affected by

98 the rainy cloud.

99 Besides meteorological conditions, dust size distribution can also influence its

100 lifetime and consequently the export efficiency of African dust towards the Amazon

101 Basin. Previous studies have observed that volume/mass fractions of coarse mode dust

102 particles, giant particles in particular, tend to be reduced along the transport due to

103 higher gravitational settling velocities (Ryder et al., 2018; Ryder et al., 2013b; Ryder

104 et al., 2013a; Van Der Does et al., 2016). Moreover, the optical properties of mineral

105 dust are also strongly size dependent, especially for those in sub-micron range (Liu et

106 al., 2018; Di Biagio et al., 2019; Ysard et al., 2018). For instance, Ryder et al. (2013a)

107 reported a loss of $60 - 90\%$ of particles $> 30$ µm in size 12 h after uplift and

108 consequently an increase in the single scattering albedo from 0.92 to 0.95 associated

109 with the change in the size distribution of dust aerosols. Therefore, the size

110 distribution of dust particles is a key factor determining the efficiency of dust

111 transport and consequently the environmental and climate effect of the mineral dust

112 downwind (Mahowald et al., 2011a; Mahowald et al., 2011b).

113 It is worth mention that the LRT events bring not only mineral dust into the Amazon

114 Basin but also biomass burning aerosols from Africa as well as sea spray aerosols

115 (Wang et al., 2016; Holanda et al., 2020; Andreae et al., 1990; Talbot et al., 1990;

116 Ansmann et al., 2009; Baars et al., 2011), making it challenging to have a quantitative

117 assessment of the impact of African dust on the Amazon Basin. So far, a few studies

118 have attempted to quantify the impact of the LRT of African dust over the Amazon

119 Basin, but mainly focus on dust deposition only (e.g. Yu et al., 2015a; Ridley et al.,

120 2012; Yu et al., 2019). Estimates of annual dust deposition and dust deposition rates

121 into the Amazon Basin exhibit a wide range (7.7-50 Tg a$^{-1}$ and 0.8-19 g m$^{-2}$ a$^{-1}$,





respectively), attributed to the application of different methods and the intrinsic
uncertainties associated with each method (Kok et al., 2021; Yu et al., 2015b;
Kaufman, 2005; Swap et al., 1992). For example, the results based on Cloud-Aerosol
Lidar and Infrared Pathfinder Satellite Observations (CALIPSO) is subject to the
uncertainty associated with the Cloud-Aerosol Lidar with Orthogonal Polarization
(CALIOP) extinction, vertical profile shape, dust discrimination, diurnal variations of
dust transport as well as the below-cloud dust missed by CALIOP (Yu et al., 2015a).
While models could be considered as a useful tool to comprehensively assess the
transatlantic transport of African dust towards the Amazon Basin and the consequent
impact over the Amazon Basin, there exist considerable differences among model
results, attributed to the uncertainties associated with the dust parameterization in the
model, including emission schemes, dust size distribution, dust deposition, etc (Kim
et al.,2014; Huneeus et al., 2011; Mahowald et al., 2014). Observational constraints
on the modelling results along the transport from source regions to receptor regions
are thus necessarily needed to demonstrate the model performance and to accomplish
a better evaluation of factors controlling the variability in the LRT of African dust and
its overall impact over the Amazon Basin assessment.
Here, we present a detailed multiyear simulation of the export of Africa dust across
the Atlantic and the impact over the Amazon Basin (around $8.8 \times 10^6$ km$^2$, see
Figure 1 for defined area) during $2013 - 2017$ with the GEOS-Chem (chemical
transport model). The aims of this study are: (1) to evaluate the model performance
regarding the simulation of dust aerosols including the particle mass size distribution
(PMSD), optical properties, mass concentrations as well as the trans-Atlantic transport
towards the Amazon Basin; (2) to analyze factors controlling the export of African
dust towards the Amazon Basin; and (3) to give a comprehensive examination of the
impact of African dust over the Amazon Basin, including aerosol concentrations,
AOD and nutrient inputs.

**2. Methodology**



## 2.1 GEOS-Chem model

In this study we use the GEOS-Chem model version 12.0.0 (www. geos-chem.org) to perform a global aerosol simulation with a horizontal resolution of $2° \times 2.5°$. The GEOS-Chem is driven by assimilated meteorological data GEOS-FP from the NASA Global Modeling and Assimilation Office (GMAO) (Lucchesi, 2013) with a native horizontal resolution of $0.25° \times 0.3125°$, which is then degraded to $2° \times 2.5°$ for the input to GEOS-Chem. We initialize the model with a 1-year spin-up followed by an aerosol simulation from 2013 to 2017.

The aerosol simulation is an offline simulation for aerosol tracers including mineral dust in four size bins, sea salt in fine ($\leq 1$ μm in diameter) and coarse ($> 1$ μm in diameter) modes, black carbon (BC), organic aerosols (OA) and sulfate-nitrate-ammonium aerosols. Aerosol optical properties used for aerosol optical depth (AOD) calculation are mainly based on Global Aerosol Data Set (Koepke et al., 1997), with modifications in aerosol size distributions (Jaeglé et al., 2011; Drury et al., 2010; Wang et al., 2003a; Wang et al., 2003b), hygroscopic growth factor of organic aerosols (Jimenez et al., 2009), and the refractive index of dust (Sinyuk et al., 2003). AOD in the model is then calculated online at selected wavelengths assuming lognormal size distributions of externally mixed aerosols and accounts for hygroscopic growth (Martin et al., 2003).

Wet deposition in GEOS-Chem, based on the scheme of Liu et al. (2001), accounts for scavenging in both convective updrafts and large-scale precipitation. Further updates by Wang et al. (2011) are also applied, accounting for ice/snow scavenging as well as the impaction scavenging in convective updrafts. Dry deposition in the model follows the standard resistance-in-series scheme by Wesely (2007), accounting for turbulent transfer and gravitational settling (Wang et al., 1998; Zhang et al., 2001).

## 2.2 Observations

The particle volume size distribution (PVSD) and AOD daily data (at wavelength of 675 nm) from AERONET level 2.0 (aeronet.gsfc.nasa.gov/new_web/download_all_v3_ aod.html, last access on Jun. 22,


2021(Dubovik et al., 2002)) during the year of 2013 − 2017 is used in the study to
evaluate dust emissions and its PMSD over the source regions in Africa. The PVSD
data provided by AERONET is a column-integrated aerosol volume size distribution
and with a size range of 0.05 − 15.0 μm. For comparison with model results, the
PVSD data is converted to PMSD using the same densities as in the model. In
addition, to minimize the influence of aerosols other than dust, only data dominated
by coarse aerosol (contribution of fine aerosol to total aerosol volume < 3% or
simulated dust contribution > 95%) is used for the comparison.
The study also uses observed PMSD over central Sahara during Fennec Campaign
(africanclimateoxford.net/projects/fennec/, last access: 22 June 2021) for the
comparison with AERONET and our model results. Aiming to investigate dust
microphysical and optical properties, 42 profiles of size distribution (0.1 − 300 μm in
diameter) over both the Sahara and the Atlantic Ocean, were sampled from in-situ
aircraft measurements during Fennec campaign. For more detailed description of the
aircraft measurements, readers are referred to Ryder et al. (2013a).
In addition, level 3 daily AOD (at wavelength of 550 nm) data from the moderate
resolution imaging spectroradiometers (MODIS) installed on Terra and Aqua
platforms (https://ladsweb.modaps.eosdis.nasa.gov/archive/allData/61/, last access: 22
June 2021) is applied in the study to evaluate the trans-Atlantic transport of dust
plumes from Africa toward Amazon Basin. For comparison, we degraded the original
horizontal resolution of MODIS data (1° × 1°) to 2° × 2.5°, consistent with the model
outputs.
Finally, long-term aerosol measurements at the Amazon Tall Tower Observatory
(ATTO, 59.0056° W, 2.1459° S, marked in Figure 1), located in central Amazon Basin
are used in the study to evaluate the influence of the long-range transport of African
dust over the Amazon Basin. The ATTO site was established in 2012 for the long-term
monitoring of climatic, biogeochemical, and atmospheric conditions in the Amazon
rainforest. Detailed description of the site and the measurements there could be found
in Andreae et al. (2015). In this study, we only use the particle number size



distribution from a Optical Particle Sizer (OPS, TSI model 3330; size range of 0.3 −
10 μm in diameter, divided into 16 size bins) and a Scanning Mobility Particle Sizer
(SMPS, TSI model 3080, St. Paul, MU, USA; size range of 10 − 430 nm in diameter,
divided into 104 size bins) over the period from 2014 to 2016. The number size
distribution is converted to mass concentrations assuming spherical particles with a
constant density of 1.5 g cm$^{-3}$ (Pöschl et al., 2010). In addition, daily PM$_{10}$ mass
concentrations during wet season (from January to April) in 2014 measured at
Cayenne, French Guiana (4.9489° N, 52.3097° W, located in the northeast coast of the
Amazon Basin, marked in Figure 1, https://doi.org/10.17604/vrsh-w974) are also used
in this study to further evaluate the model performance regarding the trans-Atlantic
transport of African dust toward the Amazon basin. The measurement at Cayenne site
is carried out on a cooperative basis by personnel of ATMO-Guyane, a non-profit
organization (https://www.atmo-guyane.org/qui-sommes-nous/statuts/). The PM$_{10}$
samples are measured by a Taper Element Oscillating Microbalance (TEOM, model
1400 series, ThermoFisher Scientific) and then are returned to Miami for analysis
(Prospero et al., 2020). Readers are referred to Prospero et al. (2020) for detailed
description of the site and the data.

**3 Dust emissions and size distribution**

**3.1 Dust emissions**

The emission of mineral dust is based on the dust entrainment and deposition (DEAD)
mobilization scheme of Zender et al. (2003) in the GEOS-Chem model (Duncan
Fairlie et al., 2007). Figure 1 shows the spatial distribution of simulated dust
emissions and Table 1 lists seasonal and annual dust emissions in northern Africa
(17.5° W − 40° E, 10° N − 35° N) for the period of 2013 − 2017. Simulated annual
dust emission from northern Africa is 0.73 ± 0.12 Pg a$^{-1}$, contributing more than 70%
of the global dust emission (0.99 ± 0.20 Pg a$^{-1}$). The result is in the range of 0.42 −
2.05 Pg a$^{-1}$ reported by Kim et al. (2014), who evaluated five AeroCom II global
models regarding the dust simulation over similar regions.



There exists a strong seasonality in the dust emission from northern Africa, peaking in
spring and winter (up to 1.2 Pg $a^{-1}$) and reaching the minimum in fall (around 0.33 Pg
$a^{-1}$) in general. Previous studies have also pointed out that dust emissions over
different African regions show distinct variations (Bakker et al., 2019; Shao et al.,
2010), attributed to differences in geographical properties (Knippertz et al., 2007),
vegetation cover (Mahowald et al., 2006; Kim et al., 2017), and meteorological
mechanisms on a local scale (Alizadeh-Choobari et al., 2014; Wang et al., 2017;
Roberts and Knippertz, 2012). Consequently, there exists substantial seasonal change
in dust source regions. For instance, during boreal winter, the Bodélé Depression in
northern Chad is found to be the most active triggered by the Harmattan winds, while
the northwestern African dust sources become less active in contrast with the
condition in boreal summer (Ben-Ami et al., 2012; Prospero et al., 2014). Therefore,
we further analyze the emission variability over five different source regions in
northern Africa (Figure 1 and Table S1). On an annual basis, the contribution to total
northern African dust emission is the largest from Region A (west Sahara, 36% ±
4.0%), followed by Region D (central Sahel including Bodélé, 21% ± 4.7%), Region
B (central Sahara, 13% ± 2.6%), Region C (east Sahara, 12% ± 1.0%), and Region E
(west Sahel, 6.5% ± 0.64%). The emission fluxes, however, are the most intensive in
Region D, up to 11 ± 2.1 g $month^{-1}$ $m^{-2}$ and are generally below 5 g $month^{-1}$ $m^{-2}$ over
the other regions. Concerning the seasonality, higher dust emission tends to occur in
boreal spring and winter, with the largest emission flux of 19 ± 4.7 g $month^{-1}$ $m^{-2}$ from
Region D. As shown in Figure 2 and S1, the emissions peak in boreal spring for
Region A, B and C, but in winter for Region D and E. There is also a secondary peak
in summer emissions for Region E. Correlation analysis between dust emissions and
meteorological variables suggests that the seasonality is mainly driven by high surface
wind speeds (with $r$ of 0.79-0.96 and 0.68-0.97 for the 75[th] and 95[th] percentiles of
wind speeds, respectively). Apparent negative correlation is also found between
precipitation (soil moisture, Figure S2) and dust emission in Region D with $r$ of -0.69
(-0.67). Similar seasonality is also reported by Cowie et al. (2014), who suggested



that the strongest dust season in winter in central Sahel is driven by strong harmattan
winds and frequent Low level Jet breakdown, and the second peak in summer in west
Sahel could be explained by the summer monsoon combined with the Sahara Heat
Low. The study also suggested the dominance of strong wind frequency in the
seasonal variation of the emission frequencies.
There is a significant decrease in the annual emission from 0.88 Pg a$^{-1}$ in 2013 to 0.56
Pg a$^{-1}$ in 2017. Similarly, studies on African dust variability at decadal and longer time
scales also reported an obviously decreasing trend in both dust emissions in Africa
and dustiness over the east mid-Atlantic in recent decades since the early 1980s
(Ridley et al., 2014; Middleton, 2019; Shao et al., 2013). Evan et al. (2016) pointed
out three periods of persistent anomalously low dust concentrations in the 1860s,
1950s and 2000s. Weather and climate drivers behind this variability include
precipitation, surface wind over northern Africa, Atlantic Multidecadal Oscillation
(AMO), North Atlantic Oscillation (NAO), the movement of ITCZ, etc. For example,
as shown in Figure S2, there is an obviously increasing trend of AMO over the period
2000 – 2015, especially from 2010 to 2015 (data available from http://www.esrl.
noaa.gov, last access on July 29, 2021 (Enfield et al., 2001)). This positive AMO
phase corresponds to higher North Atlantic sea-surface temperature (SST), and could
result in enhanced rainfall in the Sahel and consequently less African dust emissions
(Middleton, 2019). A recent study by Yuan et al. (2020) projected decreased surface
wind speeds over African dust source regions as well as more precipitation in the
Sahel region due to positive interhemispheric contrast in Atlantic SST associated with
the global warming, leading to less dust emissions and weaker westward transport.
While most regions show decline trends of dust emissions, the emission in Region D
shows a slight increase. The variation is mainly associated with surface wind speeds
(Figure S3). For instance, the 75$^{th}$ and 95$^{th}$ percentiles of wind speeds decrease by
7.0% and 9.1% in Region B but slightly increases by 2.0% and 1.4% in Region D
from 2013 to 2017. The $r$ values are in the range of 0.90 – 0.99 between annual dust
emissions and the 95$^{th}$ percentile of wind speeds over all the 5 source regions.





Significant negative correlation with *r* of -0.52 − -0.73 between annual dust emissions
and soil moisture is also found for those regions except for Region D where *r* is only -

298 0.08.

It is also worth noting that the interannual variation in dust emission is much larger
during the wet season (0.96 ± 0.25, Table 1) than on an annual basis (0.73 ± 0.12).
Moreover, while the annual emissions gradually decrease from 2013 to 2017, the
emissions during the wet season peak in 2015. The obviously different behavior
between the annual emissions and emissions during the wet season suggests that
predictions of future impact of African dust emissions over the Amazon Basin in
response to climate change should focus on the wet season rather than the annual
average, as the former is more related to the export of African dust towards the
Amazon Basin.

**3.2 Dust size distribution and its impact on the export efficiency towards the**
**Amazon Basin**
Freshly emitted dust particles are divided into four size bins in GEOS-Chem: 0.1 −
1.0 μm, 1.0 − 1.8 μm, 1.8 − 3.0 μm, and 3.0 − 6.0 μm in radius. The first size bin is
further divided into four sub-bins (0.1 − 0.18 μm, 0.18 − 0.3 μm, 0.3 − 0.6 μm, and
0.6 − 1.0 μm in radius) for the calculation of optical properties. While total dust
emissions are not affected, optical properties, atmospheric lifetime and downwind
concentrations of dust particles are sensitive to different PMSD schemes.
Table 2 shows 3 different PMSD schemes tested in this study: V12, V12_C and
V12_F. Scheme V12, which is derived based on scale-invariant fragmentation theory
(Kok, 2011) with modification in tunable parameters (Zhang et al., 2013), is a default
set in GEOS-Chem. However, this scheme has been only evaluated for US/Asian dust,
not for Africa. On the other hand, V12_C was used in older versions of GEOS-Chem
and constrained from aircraft measurements during the Saharan Dust Experiment
(Ridley et al., 2012; Highwood et al., 2003). In addition, we derived V12_F based on
the measurements during the Fennec aircraft observations also focusing on Saharan





dust. Among all the three PMSD, V12_C has the largest mass fraction in the first bin
(relatively small particles) and the lowest fraction in the last bin (large ones). In
contrast, V12_F has the most dust distributed in the last bin (~ 70%) and only a little
(around 5%) in the first bin ($0.1 - 1.0$ μm).
Simulated mass extinction efficiency (MEE, also shown in Table 2) at wavelength of
550 nm for dust particles in the first sub-bin ($0.1 - 0.18$ μm) is 3.13 $m^2$ $g^{-1}$, and
decreases to 0.16 $m^2$ $g^{-1}$ for those in the last bin ($3.0 - 6.0$ μm). The lifetime of dust
aerosols against deposition are 5.1, 2.2, 1.7 and 0.86 d in the four bins (from small to
large size), respectively. Therefore, as mentioned before, although with the same
emission, dust AOD and concentrations could vary greatly with PMSD. Here we
evaluate all the three PMSD schemes through the comparison with observed mass size
distribution of column-integrated aerosol over Africa, AOD over both Africa and the
Atlantic Ocean, as well as dust concentrations in the Amazon Basin.
Figure 3 shows the mass fractions of column-integrated aerosols retrieved from
AERONET sites compared with model results based on different PMSD schemes. The
location of the selected AERONET sites with valid data are marked in Figure 1 as
purple symbols (including asterisks and circles). The mean mass fractions for each bin
from AERONET data are 17%, 27%, 38%,17%, respectively. The comparison
indicates the model results based on V12_C agrees better with the observations. In
other words, the model results with other PMSD schemes (V12_F in particular)
greatly underestimate the mass fractions in the first bin and overestimate the mass
fractions in the last bin. During the Fennec campaign, the aircraft sampled two strong
Saharan dust outbreaks with AOD up to 1.1, which may be associated with strong
winds favoring the uplift of large particles.
Figure 4 shows the times series of daily AOD at wavelength of 675 nm during the
year of $2013 - 2017$ from both AERONET and model results. The locations of the
selected AERONET sites with valid data are shown in Figure 1 as purple circles.
Although different PSD schemes have little influence on the correlation between
AERONET and model results with most $r$ around $0.6 - 0.7$, the normalized mean bias



(NMB) has been significantly improved in V12_C, with a range of -12% − 11% (vs. -
33% − -11% and -42% − -19% for V12 and V12_F, respectively). The severe
underestimation in AOD from V12 and V12_F could be attributed to their relatively
higher dust fractions distributed in larger size bins with lower MEE.
In addition, we also compare the spatial distributions of simulated AOD over the
Atlantic Ocean with MODIS AOD (at 550 nm) averaged over 2013 − 2017 in Figure
5a-d. There is a clear decreasing trend in MODIS AOD along the trans-Atlantic
transport from Africa towards South America. Although all simulations show similar
spatial distributions with declining trends of AOD along the transport, the results from
V12_C are the most consistent with MODIS data with the highest $r$ of 0.89 and the
smallest NMB of 6.5% among the three schemes (vs. $r$ of 0.85 and 0.81 and NMB of -
13% and -19% for V12 and V12_F, respectively).
Assuming first-order removal of aerosol along the transport, we derived linear trend
lines based on the gradient of the logarithm of AOD against the distance over the
Atlantic Ocean along the transport path (AOaTP, 20° − 50° W and 5° S − 25° N,
Figure 5e). MODIS AOD decrease from 0.29 ± 0.023 near the coast of Africa to 0.17
± 0.010 at the coast of South America, with a decline rate of $0.019 \pm 0.0025$ degree$^{-1}$.
A similar decline rate is found for simulated AOD based on V12_C, decreasing from
$0.28 \pm 0.022$ to $0.16 \pm 0.013$ ($0.019 \pm 0.0029$ degree$^{-1}$). In contrast, simulations with
V12 and V12_F exhibit much lower AOD together with relatively steeper slopes of
$0.021 \pm 0.0040$ and $0.021 \pm 0.0041$, respectively. To specify the impact of different
PMSD on the export efficiency of dust aerosols towards the Amazon Basin, Figure 5f
also shows simulated dust AOD (DOD) along the transect from 20° to 50° W. The
DOD from V12_C decreases from $0.15 \pm 0.018$ to $0.049 \pm 0.088$ along the transport,
with a decreasing rate of $0.016 \pm 0.0014$ degree$^{-1}$. In contrast, DOD decreases from
$0.097 \pm 0.012$ to $0.028 \pm 0.085$ with a slope of $0.018 \pm 0.0016$ for V12 and decreases
from $0.080 \pm 0.090$ to $0.025 \pm 0.084$ with a slope of $0.017 \pm 0.0014$ for V12_F.
Lying in the trade wind belt, Cayenne has been taken as the gate of African dust.
Hence, the comparison between simulated and observed dust concentrations could



evaluate model performance in reproducing the arrival of African dust to the Amazon
Basin. As shown in Figure 6a, the simulation from V12_C shows excellent agreement
between simulated dust and observed $PM_{10}$ concentration during wet season, with $r$
around 0.85 and NMB of -39%. The correlation from the other two simulations is
similar ($r$ = 0.86), but the corresponding NMB is much larger (-57% for V12 and -
80% for V12_F). Prospero et al. (2020) did similar analysis at the Cayenne site but
concerning the data all year round. Based on the regression line between observed
concentrations of $PM_{10}$ and dust, they obtained a regional background value of $PM_{10}$
ranging from 17 to 22 $\mu g\ m^{-3}$, largely attributed to sea salt aerosol, and suggested
$PM_{10}$ values above this range as a proxy for advected dust. Consistent with their
results, the regression line between observed $PM_{10}$ and simulated dust in this study
shows a background value of $PM_{10}$ around 23 $\mu g\ m^{-3}$. The slope of the regression line
from V12_C is 1.0, also consistent with the value of 0.9 in the study of Prospero et al.
(2020), demonstrating the well performance of the model with V12_C in simulating
the trans-Atlantic transport of African dust towards the Amazon Basin. In contrast, the
regression lines from V12 and V12_F are much steeper, with the slope of 1.4 and 2.1,
respectviely.
We also compare simulated dust concentrations with observed coarse particles at
ATTO site in central Amazon in wet season during 2014 – 2016 in Figure 6b. The
correlation between observed and simulated data are similar for different PMSD
schemes with $r$ of 0.63 − 0.65. But the bias of V12_C is negligible (NMB = -0.27%)
while both V12 and V12_F tend to underestimate the coarse aerosol concentrations
with NMB of -36% and -55%, respectively. This again implies relatively higher
export efficiency of African dust aerosols towards the Amazon Basin with V12_C
associated with their relatively higher dust fractions distributed in smaller size bins.
Overall, compared with V12 and V12_F schemes, model results based on V12_C are
more consistent with the multiple observed data sets, including column-integrated
particle size distribution, AOD and surface coarse aerosol concentrations obtained
either over sources or downwind of the sources. Therefore, we use the model results



from V12_C (hereinafter referred to as model results unless noted otherwise) to
investigate the transatlantic transport of dust from Africa and its impact over the
Amazon Basin in the following sections.

**4. Transatlantic transport of African dust**
Associated with the annual oscillation of ITCZ, the dust column burden shows a steep
east-west gradient across the ocean with two major paths for different seasons (Figure
S4): one moves slightly southwest toward South America in boreal winter and spring,
and the other moves west towards the Caribbean in boreal summer and fall.
Therefore, although higher dust load over the coastal region of North Africa is found
in summer (> 500 mg m$^{-2}$), dust reaching the Amazon Basin is less than 10 mg m$^{-2}$. In
contrast, dust load over the Amazon Basin could reach up to 50 mg m$^{-2}$ in spring and
winter.
In addition to the transport path, the changes in dust column burden along the
transport towards the Amazon Basin is also sensitive to its removal rate, namely the
lifetime against deposition over the Atlantic. Assuming first-order removal of dust
aerosols, we further derived seasonal e-folding lifetime (hereinafter referred to as
lifetime) of simulated dust during 2013 − 2017, based on the logarithm of the dust
column burden against travel time over the AOaTP (Figure 7). Estimated dust lifetime
is the shortest (1.4 ± 0.098 d) in winter, followed by fall and spring (1.9 ± 0.33 d and
2.3 ± 0.31 d, respectively), while the lifetime in summer is the longest (4.2 ± 0.68 d).
The interannual variability of the lifetime is small in winter with relative standard
deviation (RSD) of 7.0%, but relatively large in fall with RSD of 17%.
The short lifetime in winter is generally associated with high deposition (with 0.18 ±
0.034 Pg a$^{-1}$ accounting 20% of the emission of Northern Africa, Table S2). As shown
in Figure 8, the spatial distribution of dust deposition is similar to that of dust burden,
again illustrating the main transport paths. The largest dust deposition flux (> 1000 ng
m$^{-2}$ s$^{-1}$) is found over the source regions in northern Africa, especially in spring and
winter, and is mainly due to dry deposition (accounting for more than 80%). As a



result, 48% – 64% of total emission in northern Africa is deposited within the source
region. The deposition flux over the AOaTP, also shows strong seasonality, with a
maximum of ~530 ng m$^{-2}$ s$^{-1}$ in winter and a minimum of ~180 ng m$^{-2}$ s$^{-1}$ in fall, and
is mainly driven by wet deposition (accounting for 76% on average). Again, although
the emissions are similar in spring and winter, the deposition flux is much larger in
winter, consistent with the relatively shorter lifetime in winter discussed above. On
the other hand, the highest dust burden (144 ± 58 mg m$^{-2}$) over the AOaTP is found in
summer mainly associated with its longer lifetime, followed by 127 ± 24, 98 ± 35, and
57 ± 20 mg m$^{-2}$ in winter, spring and fall, respectively. The deposition over the
AOaTP only accounts for 7.7% of total emission in northern Africa in Spring, in
contrast to 20% in in winter.
**5. The influence of African dust over the Amazon Basin**
**5.1 Surface aerosol concentrations and AOD**
Figure 9 shows the time series of observed and simulated aerosol mass concentrations
at ATTO in January − June for the period of 2014 − 2016. Observed mean
concentration in wet season is 9.3 ± 7.6 μg m$^{-3}$, of which 83% is from coarse aerosol
(7.7 μg m$^{-3}$), while simulated concentration is 11 ± 6.7 μg m$^{-3}$, with dust contribution
of 65% (7.2 μg m$^{-3}$). The slight model bias could be to some extent explained by the
difference in background concentrations (1.9 and 5.1 μg m$^{-3}$ for the observation and
model data, respectively). During the wet season, observed coarse aerosol
concentrations frequently exceed 9 μg m$^{-3}$, and could be up to 50 μg m$^{-3}$. Most of
observed peaks are found in February – March of 2014 and 2016, and in February −
April of 2015. The high correlation (*r* of 0.52 − 0.71) between observed coarse
aerosols and simulated dust concentrations suggests that observed strong variation in
coarse aerosols is mainly driven by the influence of dust.
The dust peaks are generally associated with large dust emission and/or efficient
trans-Atlantic transport (e.g. relatively longer lifetime). For example, the relatively
higher dust concentrations in the wet season of 2015 (except for February) are
generally associated with higher emissions (1.2 − 1.5 Pg a$^{-1}$) compared with the year



of 2014 and 2016 (0.68 – 1.0 Pg a$^{-1}$, see Table S3). On the other hand, although
emissions in February 2016 (0.95 Pg a$^{-1}$) is slightly lower than those in February 2014
(1.2 Pg a$^{-1}$), the relatively longer lifetime (1.7 d vs. 1.5 d) may help explain the high
dust concentrations during that period. It should be noted that the lifetime estimated
here represents the export efficiency averaged over a relatively large domain and
long-time scale (e.g. one month). Besides, the influence of African dust on the ATTO
site is also subject to the variations of transport paths and precipitation fields.
Over the whole Amazon Basin, simulated average surface dust concentrations in the
wet season of 2013 − 2017 are 5.7 ± 1.3 µg m$^{-3}$, with maxima over 15 µg m$^{-3}$ in the
northeast corner of rainforest and a decreasing trend towards southwestern direction
(Figure 10). The dust contribution to surface aerosol concentrations is 47% ± 5.0%
(up to 70% in the north corner). The location with the largest dust contributions
slightly shifted inland compared to the spatial distribution of dust concentration. This
could be explained by higher influence of sea salt aerosols along the coast. The ATTO
site has dust concentrations around 8.1 ±1.8 µg m$^{-3}$ (accounting for 63% ± 7.9% of
total aerosol concentrations on average) in wet season, and thus could be
representative of the whole Amazon Basin. Based on single-particle analysis using a
quantitative energy-dispersive electron probe X-ray microanalysis, Wu et al. (2019)
found that aged mineral dust and sea salts account for 37 %–70 % of the super-micron
aerosol at ATTO site during the wet season, consistent with our result. The
contribution of DOD to AOD at 675 nm over most areas of the Amazon Basin (Figure
11) is in the range of 10 − 50% (26% ± 4.7% on average) during the wet season of
2013 − 2017, with maxima in the northern Amazon Basin. The dust contribution to
total AOD is relatively smaller than that to surface aerosol concentrations, mainly
because of the relatively lower MEE of dust aerosols compared to other aerosols.
**5.2 Frequency of dust events**
Figure 10c also shows the frequency of dust events when surface dust concentrations
reach the threshold of 9 µg m$^{-3}$ on daily basis as defined in Moran-Zuloaga et al.
(2018) over the Amazon Basin in the wet season of 2013 − 2017. Dust frequency





averaged over the whole region is around 18% ± 4.6% and decreases from 50 − 60%
at the northeast coast to < 1% in southern inland. The frequency of dust events at
ATTO site is around 32%, close to the median of the range. The interannual variation
of the frequency, however, has an opposite trend, with RSD gradually increasing from
10% at the northeast coast to over 100% in southern inland (36% at ATTO). During
dust events, the dust mass concentration of ATTO reaches $16 ± 2.9$ µg m$^{-3}$ (three
times as high as that over the whole wet season), accounting for around 77% ± 5.8%
of total aerosol. Similarly, under the influence of the long-range transport of Saharan
dust plumes, Moran-Zuloaga et al. (2018) observed mass concentrations $14 ± 12$ µg
m$^{-3}$ for coarse aerosol at the same site, accounting for 93% of total observed aerosol.
There also exits large difference in DOD between the whole wet season and dust
events: $0.019 ± 0.0047$ vs. $0.038 ± 0.0074$ (at wavelength of 675 nm) averaged over
the Amazon Basin. A maximum of 0.31 on a daily basis is found on 1 Mar 2016 at the
northeast corner (55° W, 4° N) of the Amazon Basin during the study period. During
dust events, dust aerosols dominate AOD (50% − 60%) over most regions of the
Amazon Basin. At ATTO site, DOD is $0.029 ± 0.0076$ and $0.054 ± 0.0074$, accounting
for 41% and 57% of AOD over the whole wet season and dust events, respectively.
The largest dust contribution (up to 84%) with DOD of 0.15 at ATTO site is found on
24 Jan 2015. Consistent with our results, previous studies by Baars et al. (2011) and
Baars et al. (2012) reported DOD (532 nm) of up to 0.18 and AOD of ~0.14 when
affected by strong influence of Saharan dust at a similar Amazon site (60° 2.3′ W, 2°
35.9′ S).

**5.3 Dust deposition and related nutrient input**
The spatial distribution of dust deposition over the Amazon Basin is also shown in
Figure 8. The mean dust deposition flux in wet season is $2.0 ± 0.35$ g m$^{-2}$ a$^{-1}$, much
higher than in dry seasons (August to November, $0.35 ± 0.16$ g m$^{-2}$ a$^{-1}$). The maximum
(2.6 g m$^{-2}$ a$^{-1}$) is found in the year 2015 due to relatively large dust emission and
efficient trans-Atlantic transport. With emission of $0.96 ± 0.25$ Pg a$^{-1}$ in wet season





(0.73 ± 0.12 Pg a⁻¹ on annual average), only 1.9% (1.4%) of African dust is deposited
into the Amazon Basin (dominated by wet deposition) while relatively large part is
deposited over the AOaTP (13% in the wet season and 14% on annual average) and
northern Africa (49% in the wet season).
Assuming mass fractions of 4.4%, 0.082%, and 1.8% for iron, phosphorus, and
magnesium respectively in the African dust (Bristow et al., 2010; Chiemeka et al.,
2007), we derive deposition fluxes of 88 ± 15, 1.6 ± 0.29 mg m⁻² a⁻¹ and 36 ± 6.3 mg
m⁻² a⁻¹ for iron, phosphorus and magnesium respectively into the Amazon rainforest
during the wet season and 52 ± 8.7, 0.97 ± 0.16 and 21 ± 3.6 mg m⁻² a⁻¹ on annual
average (Figure 12). It should be noted that there exits large spatial variation of
nutrient input into the Amazon Basin associated with the patterns of dust burden and
dust deposition flux. The deposition flux decreases from over 70 mg m⁻² a⁻¹ at
northeast coast to less than 7 mg m⁻² a⁻¹ in inland for magnesium and decreases
from > 9 mg m⁻² a⁻¹ at northeast coast to less than 1 mg m⁻² a⁻¹ in southwestern Basin
for phosphorus during the wet season. Similarly, the deposition flux of iron during the
wet season decreases from over 500 mg m⁻² a⁻¹ at northeast coast to less than 15 mg
m⁻² a⁻¹ in the southwest and is above 50 mg m⁻² a⁻¹ in most of the Amazon Basin. It
seems that the nutrient input from Africa dust may play a significant role in the
northeastern part of the Amazon Basin, not in the southwest.
Table 3 summarized the estimates of dust and associated phosphorus deposition into
the Amazon Basin from previous studies. The estimated fluxes of dust and associated
phosphorus deposition are in the range of 0.81 – 19 g m⁻² a⁻¹ and 0.48 – 16 mg m⁻² a⁻¹.
The large range is mainly driven by the high values (19 g m⁻² a⁻¹ and 16 mg m⁻² a⁻¹ for
dust and associated phosphorus, respectively) from the study of Swap et al. (1992).
Based on observations during storm events and dust climatology, the study estimated
dust importation into the northeastern basin, which is most subject to the intrusion of
African dust. Besides the discrepancy in defined regions, the wide range could also be
partly explained by the application of different methods and associated intrinsic
uncertainties as mentioned in the Introduction. For instance, the estimates from Swap





et al. (1992) are mainly based on 1-month field measurements at three sites located in
the northeastern basin. Assumption about air exchange rate across the coast to the
basin, duration of dust storms as well as dust concentrations contained in the dust
plumes had to been made to extrapolate the dust deposition into the Amazon Basin.
Similarly, bias could also arise from insufficient observations available to constrain
models or satellite retrievals. Additional uncertainty may also stem from the
assumption about the P mass fraction, ranging from 0.07% to 0.108%. Our results are
similar to the finding of Prospero et al. (2020), which has also been constrained by the
observation at Cayenne.
According to Salati and Vose (1984), the total amount of phosphorous and magnesium
is 21.6 g m$^{-2}$ and 29.8 g m$^{-2}$, respectively, in the ecosystem of the Amazon Basin (14.7
and 2.3 g m$^{-2}$ respectively in the soil). On the other hand, Vitousek and Sanford
(1986) reported a loss of $0.8 - 4$ mg m$^{-2}$ a$^{-1}$ for phosphorus and 810 mg m$^{-2}$ a$^{-1}$ for
magnesium in Brazilian ecosystem to surface waters. Estimated nutrient input from
African dust in our study accounts for 0.011% and 1.6% of total phosphorous and
magnesium in the soil over the Amazon Basin during the wet season (0.0066% and
0.91% on annual average), and could almost compensate the hydrologic losses of
phosphorous in Brazilian forest ecosystem. Similarly, Abouchami et al. (2013)
pointed out that most of the Amazonian rainforest is a system with an internal
recycling of nutrients. But the extra influx of nutrients from African dust might
account for a significant portion of the net ouflux, i.e. dissolved discharge of nutrients
into rivers. Keep in mind that the estimates of nutrients influx are subject to the
uncertainties in the estimates of dust flux as well as the mass fractions of nutrients
containted in the dust. In addition, marine aerosols and biomassburning aerosols
mixed with the LRT of African dust may also play a role for certain essential nutrients
(Prospero et al., 2020; Abouchami et al., 2013).

**6. Conclusion**
In this study, we use the GEOS-Chem model with optimized particle mass size





distribution (PMSD) of dust aerosols to investigate the influence of the export of
African dust towards the Amazon Basin during 2013 – 2017. The model performance
is constrained by multiple datasets obtained from AERONET, MODIS, as well as
Cayenne and ATTO sites in the Amazon Basin, including particle size distribution
over Africa, aerosol optical depth (AOD) over Africa and the Atlantic Ocean as well
as coarse and total aerosols concentrations in the Amazon Basin.
Simulated dust emission from northern Africa is $0.73 \pm 0.12$ Pg a$^{-1}$, accounting for
more than 70% of global dust emission. There exists a strong seasonality in dust
emission with peaks in spring or winter, which varies with source regions and is
mainly driven by high surface wind speeds. It is worth noting that no consistent
decline is found for dust emission during the wet season, when the export of African
dust towards the Amazon Basin is more efficient driven by the southward movement
of ITCZ.
In addition to the transport path associated with the oscillation of ITCZ, the export
efficiency of African dust towards the Amazon basin is sensitive to the removal of
dust aerosol along the trans-Atlantic transport, which also depends on assumed PMSD
of dust aerosols in the model. The optimized PMSD in this study well captures
observed AOD regarding both the mean value as well as the decline rate of the
logarithm of AOD over the Atlantic Ocean along the transport path (AOaTP), while
the other two PMSD schemes tend to overestimate the decline rate by 11% and
underestimate the mean value by up to ~40%. The study further estimates the e-
folding lifetime of dust aerosols along the trans-Atlantic transport based on the
logarithm of the dust column burden against travel time over the AOaTP. The shortest
lifetime (1.4 d) is found for winter associated with high deposition flux, while the
highest dust burden over the AOaTP is found in summer mainly associated with its
longer lifetime (4.2 d).
Simulated surface dust concentration averaged over the whole Amazon Basin is $5.7 \pm$
$1.3$ µg m$^{-3}$ during the wet season of 2013 – 2017, contributing $47\% \pm 5.0\%$ to total
surface aerosols. Observed dust peaks at the ATTO site are generally associated with





large dust emission and/or efficient trans-Atlantic transport. The frequency of dust
events is 18% ± 4.6% averaged over the Amazon Basin and up to 50% − 60% at the
northeast coast. During the dust events, DOD is around 0.038 and dominate total
AOD over most of the Amazon Basin. Associated with the deposition of African dust,
the study estimated annual inputs of $52 \pm 8.7$, $0.97 \pm 0.16$ and $21 \pm 3.6$ mg m$^{-2}$ a$^{-1}$ for
iron, phosphorus and magnesium into the Amazon rainforest, which may well
compensate the hydrologic losses of nutrients in the forest ecosystem.

Acknowledgements.
This work is supported by the National Natural Science Foundation of China (41907182,
41877303, 91644218), the National key R&D Program of China (2018YFC0213901), the
Fundamental Research Funds for the Central Universities (21621105), the Guangdong
Innovative and Entrepreneurial Research Team Program (Research team on atmospheric
environmental roles and effects of carbonaceous species: 2016ZT06N263), and Special Fund
Project for Science and Technology Innovation Strategy of Guangdong Province
(2019B121205004). We acknowledge the support by the Instituto Nacional de Pesquisas da
Amazônia (INPA). We would like to thank all people involved in the technical, logistical, and
scientific support within the ATTO project.

*Financial support.* This work is supported by the National Natural Science Foundation of
China (41907182, 41877303, 91644218), the National key R&D Program of China
(2018YFC0213901), the Fundamental Research Funds for the Central Universities
(21621105), the Guangdong Innovative and Entrepreneurial Research Team Program
(Research team on atmospheric environmental roles and effects of carbonaceous species:
2016ZT06N263), and Special Fund Project for Science and Technology Innovation Strategy
of Guangdong Province (2019B121205004). For the operation of the ATTO site, we
acknowledge the support by the Max Planck Society (MPG), the German Federal Ministry of
Education and Research (BMBF contracts 01LB1001A, 01LK1602B, and 01LK2101B) and
the Brazilian Ministério da Ciência, Tecnologia e Inovação (MCTI/FINEP contract





01.11.01248.00), the Amazon State University (UEA), FAPEAM, LBA/INPA, FAPESP -
Fundação de Amparo à Pesquisa do Estado de São Paulo, grant number 2017/17047-0, and
SDS/CEUC/RDS-Uatumã. XW acknowledges the financial support of China Scholarship
Council (CSC). MP acknowledges the financial support by the Max Planck Graduate Center
with the Johannes-Gutenberg University, Mainz.

Competing interests.
Hang Su and Yafang Cheng are members of the editorial board of Atmospheric Chemistry and
Physics.

Data availability.
OPS data used in this study could be found at https://www.attodata.org/. Other datasets are
available upon request.

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

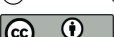


**Table 1.** Annual and seasonal dust emissions (Pg a$^{-1}$) in northern Africa (17.5° W –
40° E, 10° N – 35° N)[a].

| Year | Spring | Summer | Fall | Winter | Annual (Wet season) |
|---|---|---|---|---|---|
| 2013 | 1.2 | 0.77 | 0.48 | 1.0 | 0.88 (1.1) |
| 2014 | 0.83 | 0.84 | 0.51 | 0.91 | 0.77 (0.89) |
| 2015 | 1.2 | 0.46 | 0.33 | 1.1 | 0.77 (1.3) |
| 2016 | 0.82 | 0.52 | 0.37 | 0.89 | 0.65 (0.86) |
| 2017 | 0.68 | 0.38 | 0.47 | 0.70 | 0.56 (0.63) |
| Mean±std[b] | 0.95±0.24 | 0.59±0.20 | 0.43±0.078 | 0.92±0.15 | 0.73±0.12 (0.96±0.25) |

[a] Spring: March – May; Summer: June – August; Fall: September – November; Winter: January,
February, and December; Wet season: January – April
[b] standard deviation


**Table 2.** Mass fractions (%) of dust emitted in each bin for different particle mass size
distribution (PMSD) schemes tested in GEOS-Chem.

| Scheme | bin 1 | | | | bin 2 | bin 3 | bin 4 |
|---|---|---|---|---|---|---|---|
| | sub-bin 1 | sub-bin 2 | sub-bin 3 | sub-bin 4 | | | |
| | (0.1 – 0.18)[a] | (0.18 – 0.3)[a] | (0.3 – 0.6)[a] | (0.6 – 1.0)[a] | (1.0 – 1.8)[a] | (1.8 – 3.0)[a] | (3.0 – 6.0)[a] |
| | (3.1)[b] | (4.3)[b] | (2.7)[b] | (0.96)[b] | (0.45)[b] | (0.27)[b] | (0.16)[b] |
| V12 | 7.7 | | | | 19.2 | 34.9 | 38.2 |
| | 0.7 | 3.32 | 24.87 | 71.11 | | | |
| V12_C | 12.2 | | | | 25.3 | 32.2 | 30.2 |
| | 6 | 12 | 24 | 58.00 | | | |
| V12_F | 5.5 | | | | 11.9 | 15.6 | 67 |
| | 3.9 | 8.06 | 43 | 45.04 | | | |

[a] size range in radius (μm) for each bin
[b] mass extinction efficiency (MEE) at wavelength of 550 nm in unit of m$^2$ g$^{-1}$ for dust particles in
each bin in the GEOS-Chem model











**Table 3.** Estimates of annual dust and associated phosphorus deposition into the

Amazon Basin.

| Methods | Dust deposition | | P deposition | | References |
|---|---|---|---|---|---|
| | total | flux | total | flux | |
| | ($Tg\ a^{-1}$) | ($g\ m^{-2}\ a^{-1}$) | ($Tg\ a^{-1}$) | ($mg\ m^{-2}\ a^{-1}$) | |
| CESM2 | $10 \pm 2.1$ | n/a | $0.0077 \pm 0.0016$ | n/a | Li et al. (2021)[a] |
| AeroCom Phase I | 7.7 | 0.81 | 0.0063 | 0.66 | Kok et al. (2021)[b] |
| MERRA-2 | 8.0 | 1.05 | 0.0062 | 0.9 | Prospero et al. (2020)[a] |
| MERRA-2, CAM | n/a | n/a | $0.011 - 0.033$ | $1.1 - 3.5$ | Barkley et al. (2019)[a] |
| GLOMAP | 32 | 1.8 | 0.019 | 1.1 | Herbert et al. (2018)[a] |
| CALIOP | $8 - 48$ | $0.8 - 5$ | $0.006 - 0.037$ | $0.7 - 3.9$ | Yu et al. (2015b)[a] |
| ECHAM5 | 30.3/11.4 | n/a | 0.025/0.0093 | n/a | Gläser et al. (2015)[b] |
| GEOS-Chem | $17 \pm 5$ | n/a | 0.014 | n/a | Ridley et al. (2012)[b] |
| MATCH | n/a | n/a | n/a | 0.48 | Mahowald et al. (2005)[a] |
| MODIS | 50 | n/a | 0.041 | n/a | Kaufman (2005)[b] |
| Field measurement | 13 | 19 | 0.011 | 16 | Swap et al. (1992)[b] |
| GEOS-Chem | $10 \pm 1.7$ | $1.2 \pm 0.20$ | $0.0085 \pm 0.0014$ | $0.97 \pm 0.16$ | This study |

*Note*. Table extracted in part from Prospero et al. (2020).
[a] The P mass fraction is 0.077% for Li et al. (2021) and Prospero et al. (2020), 0.108% for Barkley

et al. (2019), 0.088% for Herbert et al. (2018), 0.078% for Yu et al. (2015b), and 0.07% for

Mahowald et al. (2005).

[b] Assuming P mass fraction of 0.082% in dust, the same value as used in this study.




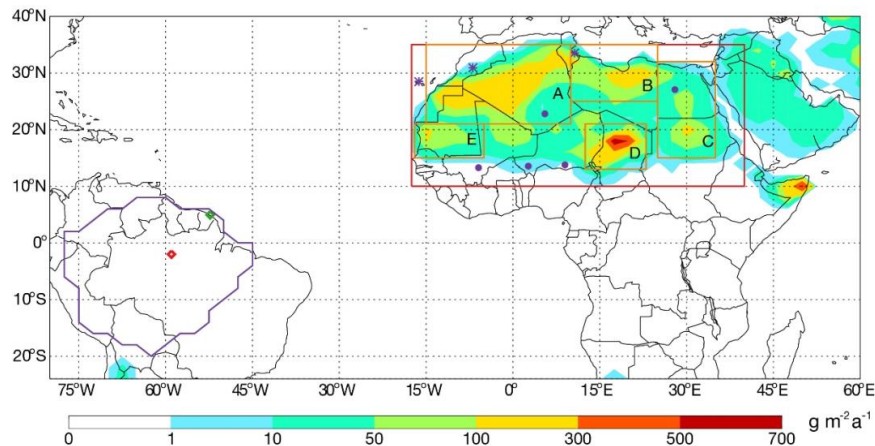

**Figure 1.** Simulated dust emissions in GEOS-Chem, averaged from 2013 to 2017. The location of AERONET sites used in Figure 3 are marked as purple symbols, of which circles represent the sites used in Figure 4. The region of the Amazon Basin is defined by purple lines. The location of Cayenne site in the northeast coast of South America and ATTO site in the central Amazon Basin are marked as green and red diamonds, respectively. The red rectangle illustrates the area of northern Africa (17.5° W – 40° E, 10° N – 35° N) and the orange rectangles shows the areas of five major source regions described in the text (A: 15° W – 10° E, 21° N – 35° N; B: 10° E – 25° E, 25° N – 35° N; C: 25° E – 35° E, 15° N – 32° N; D: 12.5° E – 23° E, 13° N – 21° N; E: 17° W – 5° W, 15° N – 21° N).




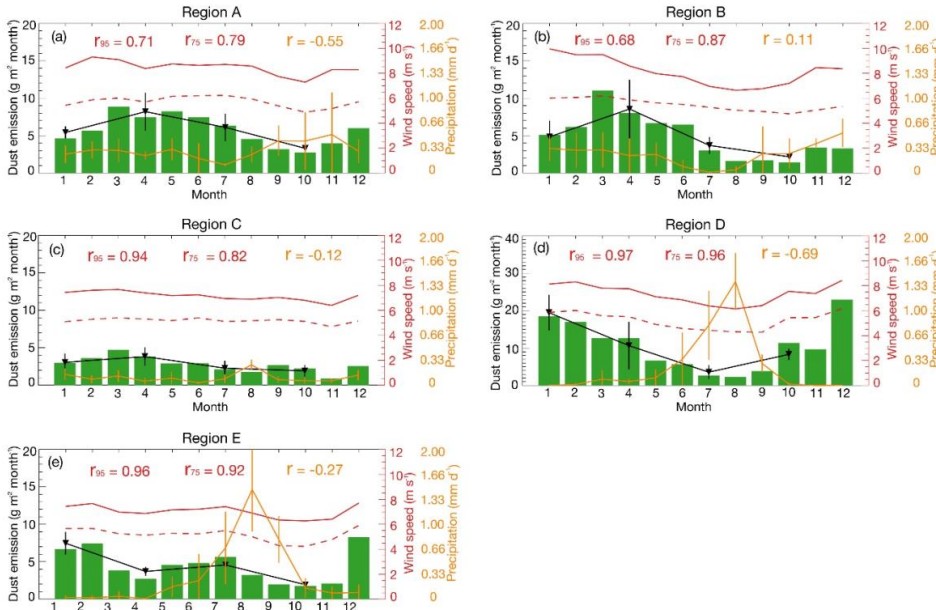

**Figure 2.** Monthly dust emission fluxes together with the 95[th] percentile hourly wind
speeds (red solid lines), the 75[th] percentile hourly wind speeds (red dotted lines) and
precipitation (yellow lines) over the five major source regions averaged from 2013 to
2017. Seasonal emission fluxes of dust are also shown as black lines. The correlation
coefficients (*r*) between the dust emission fluxes and different meteorological variables
are also shown in each panel.


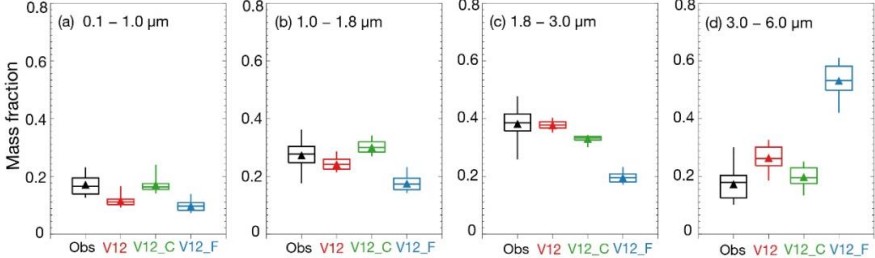

**Figure 3.** Boxplots of the mass fractions of column-integrated aerosols in the four size
bins (in radius) retrieved from AERONET sites over Africa compared with model
results based on different PMSD schemes. The triangles represent the mean values.

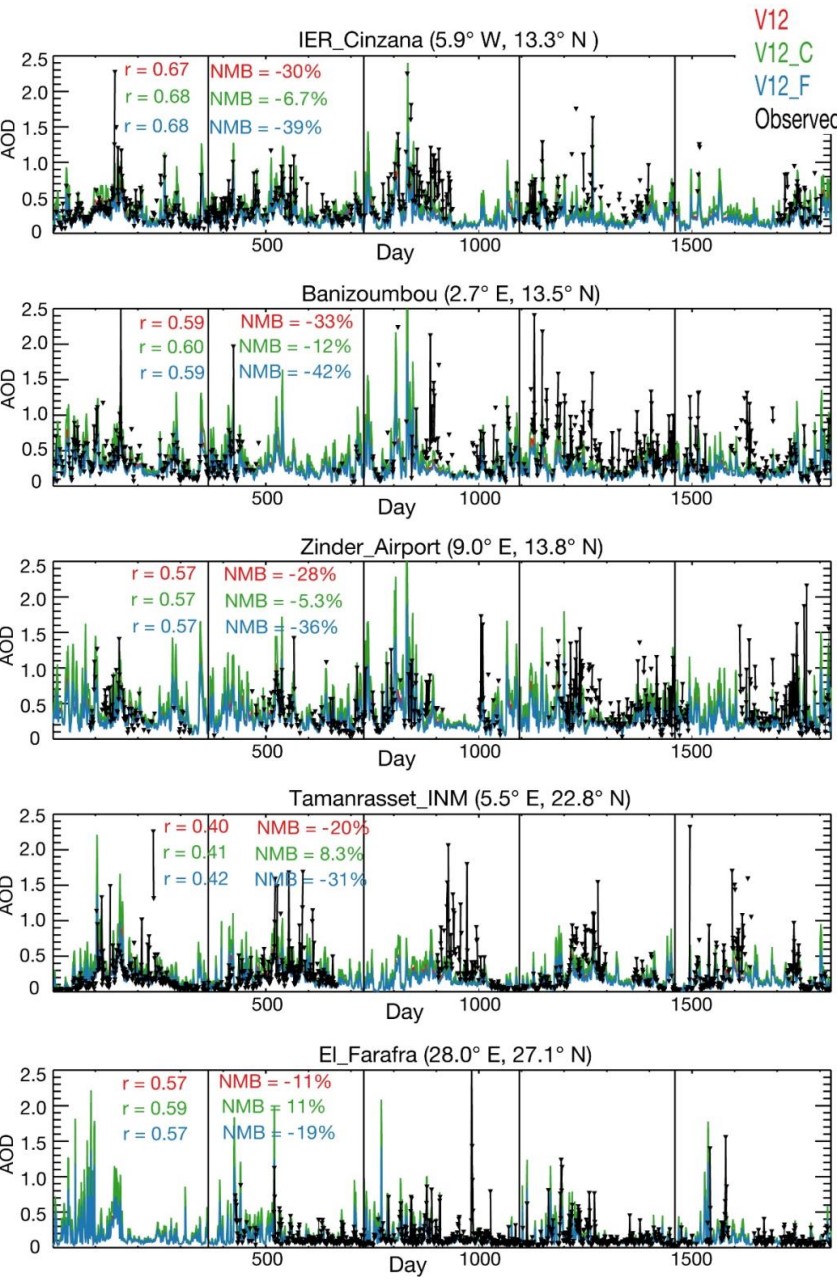

**Figure 4.** Time series of AERONET (black lines) and simulated daily AOD (at wavelength of 675 nm) during 2013 – 2017. Normalized mean bias (NMB) and correlation (*r*) statistics between the AERONET and simulated data are shown as inset.

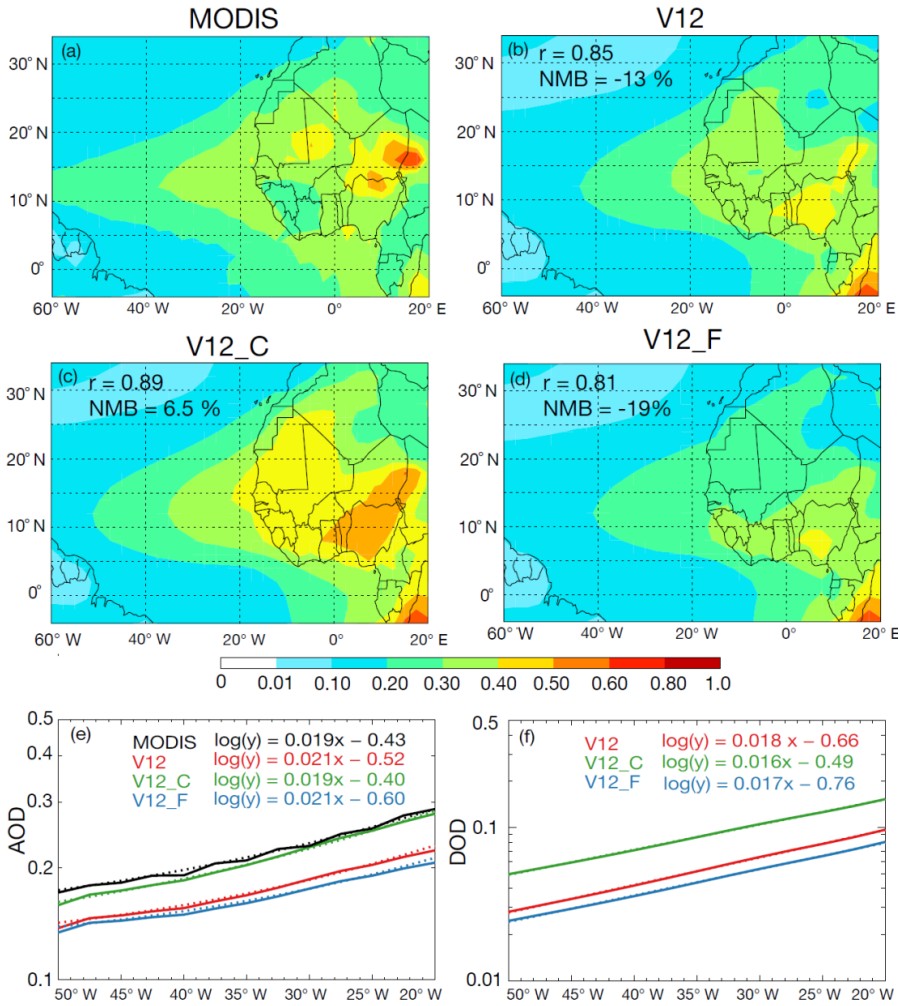

**Figure 5.** (a) – (d) Spatial distributions of observed and simulated AOD (at 550 nm) over the region of 60° W – 20° E and 10° N – 35° N averaged over 2013 – 2017. Normalized mean bias (NMB) and correlation coefficient (r) between the simulations and MODIS AOD are shown as inset. (e) MODIS (black) and simulated (color) AOD and (f) simulated dust optical depth (DOD) at 550 nm along the transect from 20° to 50° W, averaged over 5° S − 25° N for the period 2013 − 2017. The solid lines represent averaged data and the dashed lines are the logarithmic trend lines.



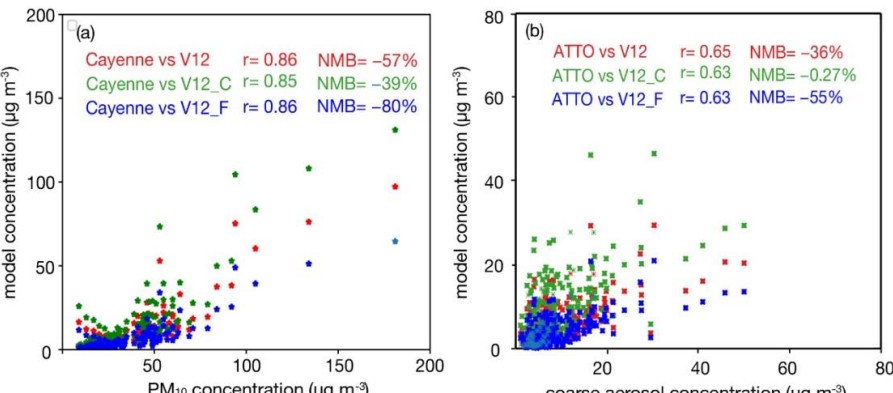

1116

**Figure 6.** Scatter plots of (a) observed $PM_{10}$ and simulated dust concentrations at Cayenne site during wet season of 2014 and (b) observed coarse aerosol ($PM_{1-10}$) and simulated dust concentrations at ATTO site during wet season of 2014-2016. Normalized mean bias (NMB) and correlation (*r*) statistics between the observation and simulation are shown as inset.

1122

1123

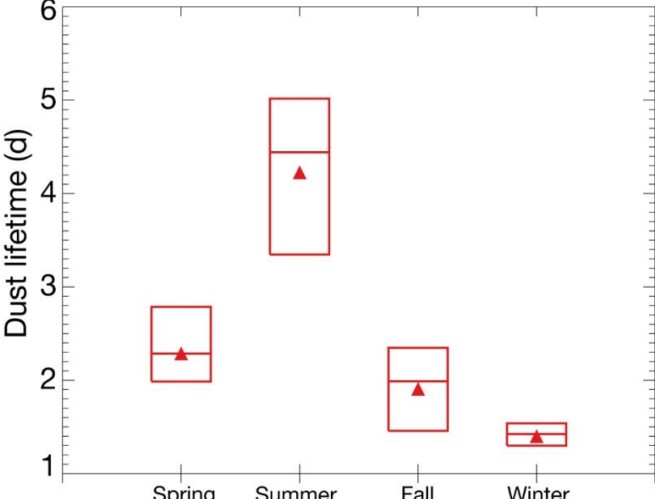

1124

**Figure 7.** Seasonal e-folding lifetime derived from the logarithm of simulated dust column burden against travel time along the transect from 20° W to 50° W averaged over 5° S − 25° N during the period of 2013 – 2017. The triangles represent the mean values, and the bottom and top sides of the boxes represent the minimums and maximums.

**Figure 8.** Simulated seasonal (left) dust deposition fluxes and (right) contribution of
wet deposition during 2013-2017. The ATTO site is marked as asterisk. The region of
the Amazon Basin is defined by purple lines in Figure 7a.

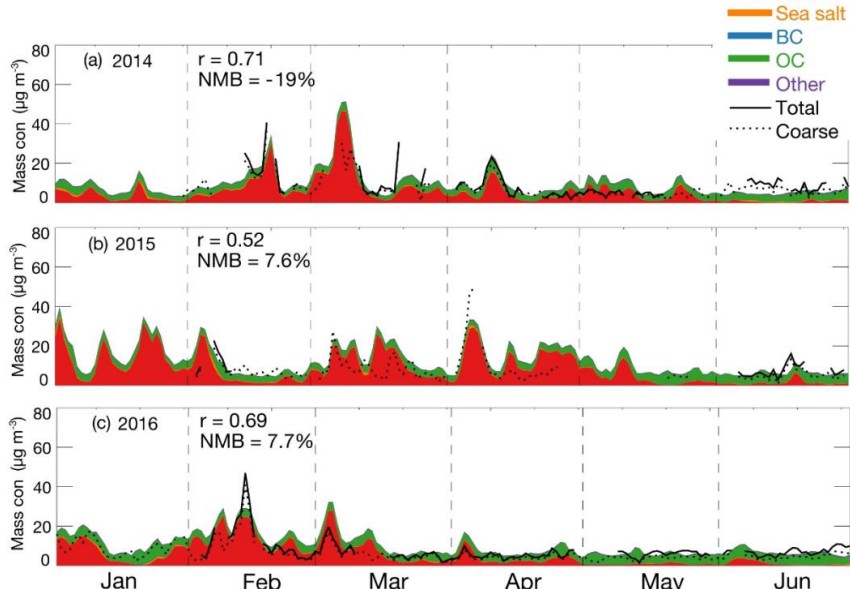


**Figure 9.** Time series of observed coarse and total aerosol mass concentrations and
simulated aerosol species concentrations at the ATTO site from January to June in (a)
2014, (b) 2015, and (c) 2016. Model results are separated into different species shown
as stacked areas. Normalized mean bias (NMB) and correlation coefficient (r) between
the observed coarse aerosols and simulated dust concentrations are shown as inset.

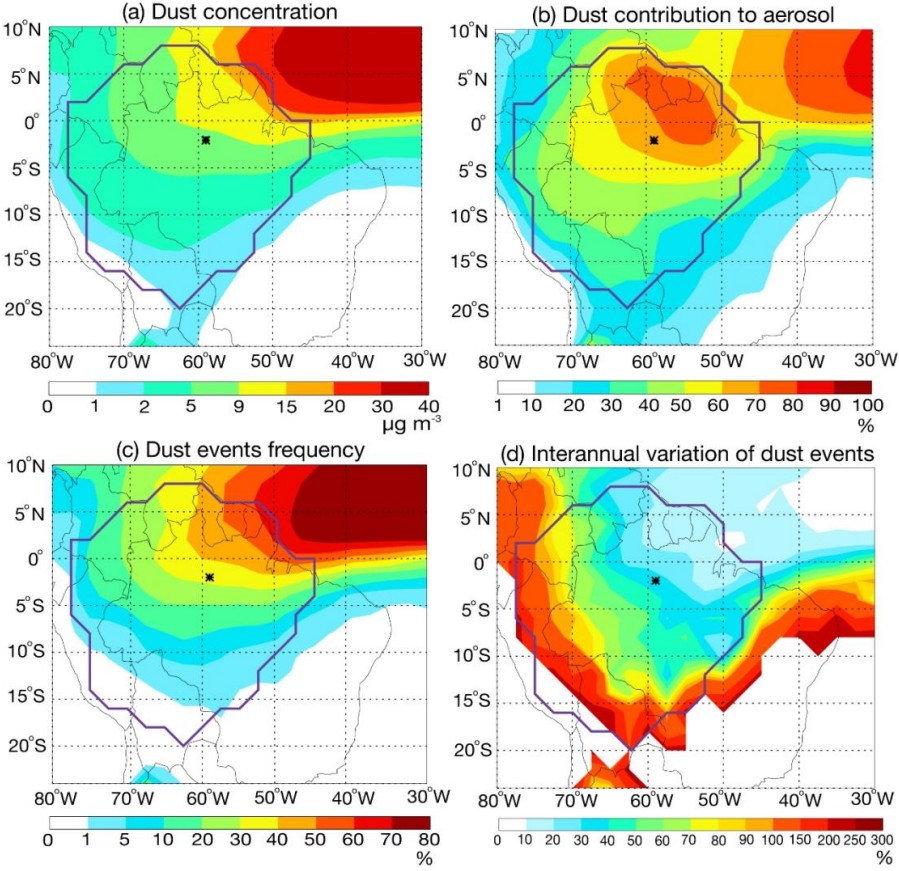

**Figure 10.** (a) simulated surface dust concentration and (b) its contribution to surface aerosol concentration over the Amazon Basin in the wet season of 2013-2017. (c) the frequency of dust events and (d) its interannual variation (namely relative standard deviation) during the same period. The location of ATTO site is marked as asterisks. The region of Amazon Basin is marked by purple lines.





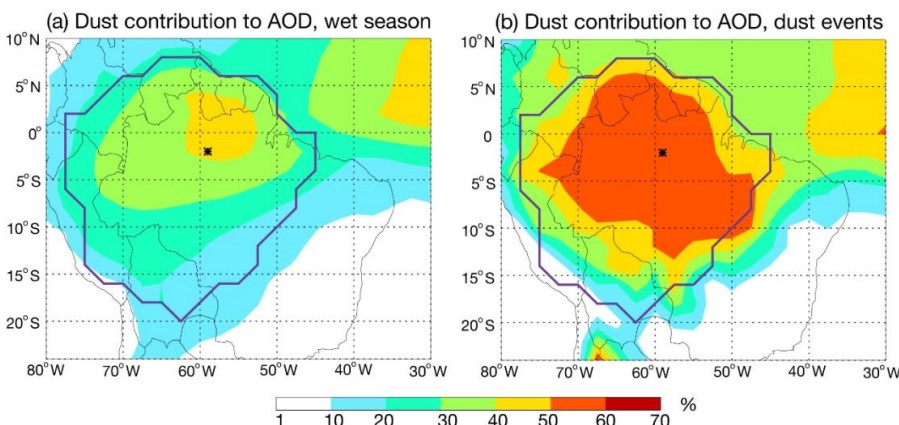

**Figure 11.** Dust contribution to total AOD at 675 nm over the Amazon Basin averaged over the (a) wet season and (b) dust events during 2013-2017. The region of Amazon Basin is marked by purple lines.

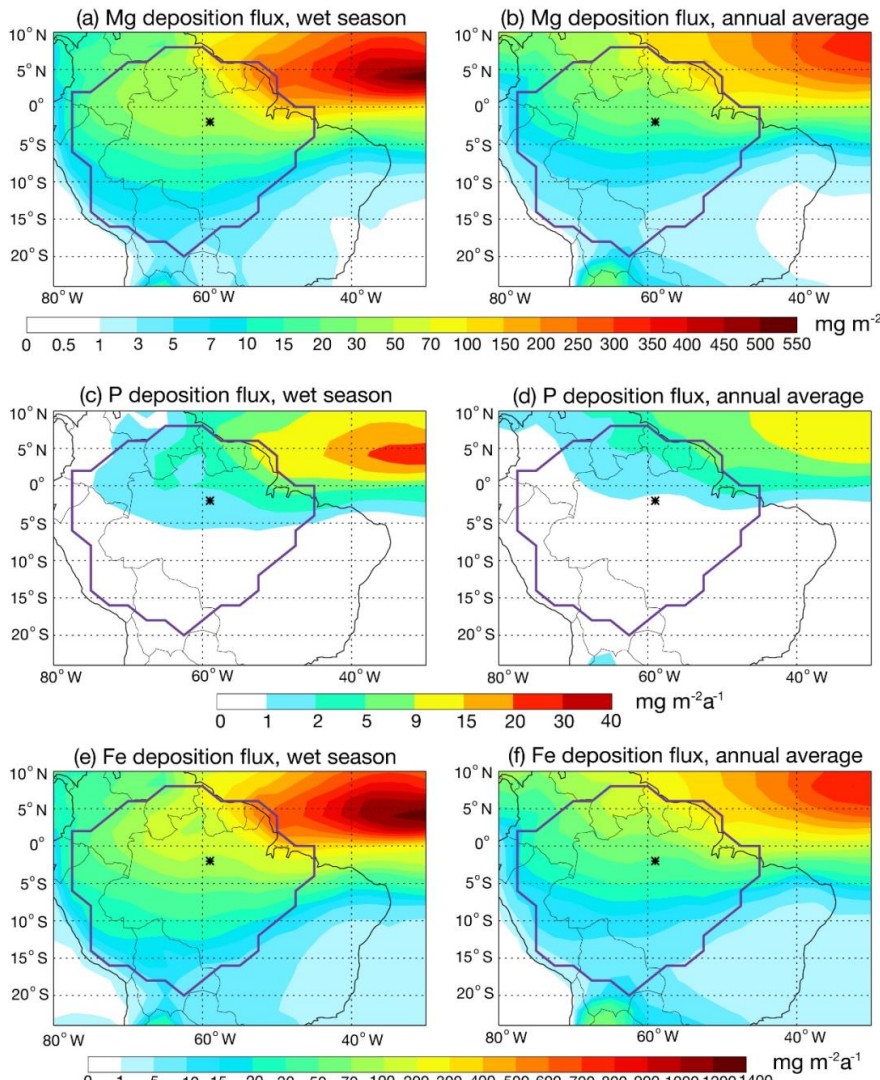

**Figure 12.** Magnesium deposition flux in (a) wet season and (b) annual averaged from 2013 to 2017. Phosphorus deposition flux in (c) wet season and (d) annual averaged from 2013 to 2017. Iron deposition flux in (e) wet season and (f) annual averaged from 2013 to 2017. The location of ATTO site is marked as asterisks. The region of Amazon Basin is marked by purple lines.