# Peer review of "Figure S1. Monthly dust emission fluxes together with the soil moisture over each region averaged from 2013 to 2017. Seasonal fluxes of dust emissions are shown as black lines. The correlation coefficients ( $r$ ) between the dust emission fluxes and soil moisture are also shown as inset."

_Atmospheric Chemistry and Physics, 2022_

## Author Comment (AC1)

**We thank the reviewers for their supportive and thoughtful comments. Our responses to the comments are provided below in a red font, with the reviewers' comments in a black italicized font.**

Review #1

*General Comments:*

*This is a very well written paper that discusses several aspects of the Saharan dust transport towards the Amazon Basin, from its origin to the impacts over the South American rainforest. The use of the chemical transport model GEOS-Chem, constrained by observations, is a very interesting approach, since it allows the assessment of regions and features not possible by real world observations, while validation against real world observations assure the general accuracy of the simulations. This paper represents a valid effort to better understand the transport and impacts of the Saharan dust.*

Thanks for the reviewer's thoughtful comments to help us improve the manuscript. We have now addressed all the concerns. Please see our detailed reply below.

1. *Some aspects about methodology and results of the comparison between AERONET observations and PMSD schemes might need some clarification. Between lines 185-187, you say that only observations dominated by coarse aerosols are used [(contribution of fine aerosol to total aerosol volume < 3%)]. But figure 3 also shows box-plots of the mass fractions of column integrated aerosols in the 0.1-1.0 micrometers size bin. In addition to that, according to figure 1, you also use one AERONET site on an island and at least two sites (in Marrocco and in Tunisia) not far from the coast. I wonder if sea salt contribution to coarse size aerosols will significantly affect the observations. And if yes, to what extent. If what the observations show are significantly affected by sea salt, extra care should be taken while drawing conclusions from the comparisons.*

*Some clarification on this aspect of the methodology and how this could affect the results could be helpful.*

Sorry for the confusion. The data screening criteria are slightly different between Fig. 3 and Fig. 4. It is more stringent for Fig. 3 as it is for the evaluation of PMSD of dust in GEOS-Chem. So it only uses data dominated by dust, namely when the contribution of dust to column-integrated aerosols mass concentrations in the model is higher than 95%. Therefore, it contains the mass fractions of column integrated aerosols in the 0.1-1.0 micrometers size bin. The influence of sea salt is limited with this screening criterion. For Fig. 4, it uses the data dominated by coarse aerosols (the contribution of fine aerosol to total aerosol volume < 3%) based on the AERONET PVSD to have more data available for the comparison between observed and simulated AOD. We have revised the text between lines 222-231 to make it more clearly: "**In addition, to minimize the influence of aerosols other than dust, only data dominated by dust (simulated dust contribution to column-integrated aerosols mass concentrations > 95%) is used for the comparison of PMSD. There are a few sites not far from the coast and could be influenced by sea salt. With the above data screening, the sea salt contribution to total aerosol mass is less than 0.5%. For the comparison of AOD, the criterion is less stringent to have more data points available and uses data dominated by coarse aerosols (the contribution of fine aerosol to total aerosol volume < 3%). This criterion does not exclude sea salt and the contribution of sea salt to AOD could be up to 30% at the Capo_Verde site (22.9° W,16.7° N) over the east of the Atlantic Ocean.**"

2. *The discussion about the dust emissions (section 3.1) feels incomplete. The winds being a major driver of the emissions is an important and interesting aspect, but it was already pointed out in several previous papers. The potential relevance of soil moisture for all regions except for region D, suggested by the significant negative correlations, is another important and interesting aspect but also a more*

*novel one, which should be more highlighted and/or discussed (e.g. in the conclusions). Regarding the winds, I expected a wider discussion on the local and synoptical meteorological aspects which result in those winds. This is briefly discussed around line 268, where the emissions from central Sahel and west Sahel are mentioned. But Region A (west Sahara), referred to as the biggest dust source, is not even mentioned.*

Thank you for the nice suggest. We now add more discussion regarding the influence of winds and soil moisture in lines 405-411: "**Fiedler et al. (2013) also found a maximum of emission flux over the Bodélé Depression in winter and the highest emission flux in spring in west Sahara. The study suggested that near-surface peak winds associated with Nocturnal Low-Level Jets is a driver of mineral dust emissions. Negative correlation between dust emissions and soil moisture has also been revealed by Yu et al. (2017) and Pierre et al. (2012), as the decreased vegetation growth in response to dry soil would result in enhanced dust emissions.**"
We also modified the conclusion to highlight the results in line 622-624: "**The correlation analysis suggests high surface wind speeds and low soil moisture as a major driver for dust emissions.**"

3. *The dust lifetime is presented in section 4, and the differences are justified mostly by dry deposition near the source and by wet deposition along the transport path. That is another interesting result, but I also feel it could have a wider discussion, especially regarding the aspects involving dry deposition. Different seasons will obviously have different meteorological and thermodynamic conditions and these different conditions will result in different structures of the dust plumes. I would expect this to be of big relevance for the dust lifetime. I would recommend the reading of "The Three-Dimensional Structure of Transatlantic African Dust*

*Transport: A New Perspective from CALIPSO LIDAR Measurements" by Liu et al. (2012), and/or related papers.*

Thank you for the insightful comment. We now had more discussion regarding dust lifetime in line 456-462: **"The seasonality in the deposition fluxes and the consequent dust lifetime depends not only on precipitation but also the vertical pathways of dust transport across the Atlantic. Dust aerosols aloft at higher altitude reach further west and have relatively longer lifetime. Significant differences in dust vertical distributions along the transport pathways have been revealed from the CALIOP measurements, which show that more dust is transported above 2km in summer while the dust layer is the shallowest in winter (Liu et al., 2012)."**

**Specific Comments:**

*1. l. 36: Pg a$^{-1}$ is an unusual notation, I would recommend using (in this line but also in the rest of the manuscript) a more common notation like Pg yr$^{-1}$*

We have modified the unit throughout the manuscript.

*2. l. 54: You wrote downwards, but I think you meant downwind?*

We have corrected the mistake.

*3. l. 96: It would be nice if you included more information about where El Djouf is located. Either "El Djouf, between Mauritania and Mali" or "El Djouf, in western Sahara".*

Thanks for the comment. We have modified the text to specify it: **"**Yu et al. (2020) argued that El Djouf**, in western Sahara,** contributes more dust...**"**

*4. l. 203: I think there should be a comma after "Amazon Basin".*

We have added the comma.

5. *l. 300: Please include units.*

   Done.

6. *l. 509: I think you meant "exists" and not "exits".*

   We have corrected the typo.

7. *l. 534: Units for the first values are missing.*

   Thanks for the comment. We have added its unit.

8. *l. 595: Maybe substitute "consistent" with "significant".*

   Thanks for the comments. The sentence is deleted in the revised manuscript.

---

## Author Comment (AC3)

**We thank the reviewers for their supportive and thoughtful comments. Our responses to the comments are provided below in a red font, with the reviewers' comments in a black italicized font.**

**Review #2**

*General Comments:*

*The paper discusses the export of African dust across the Atlantic and its impact on the Amazon Basin, using mainly the GEOS-Chem model results and a few observations. This paper discussed several aspects about the export of African dust across the Atlantic and its impact on the Amazon Basin. It provides many results and statistical numbers, which are mainly based on the GEOS-Chem model simulation and a few observations. Though it raised some interesting topics with a lot of analysis, the important scientific points are not focused and highlighted enough with strong evidence. In some places, the descriptions are unclear and not accurate enough. I suggest focusing on fewer aspects and providing stronger evidence through more observations or model sensitivity experiments. This would allow for a more interesting and well-focused study, rather than trying to cover too many aspects. In my opinion, further study of 1-2 sections in this paper can be a very interesting study and well-focused paper.*

Thanks for the reviewer's thoughtful comments to help us improve the manuscript. We have re-organized the structure of the manuscript and now have a new section (**Section 3. Model evaluation**) focusing on the model evaluation regarding dust simulation. We also add a new table (**Table 2. Summary of the observations used in this study, including the parameters, the spatio-temporal coverage, and the corresponding application in the model**) in Section 2.2 (Observations) to have a better overview of the observations used in this study. The main goal of the model evaluation in this study is to have a better PMSD scheme for freshly emitted African dust so as to better simulate the export of African dust towards the Amazon Basin. We agree with the reviewer that more observations would provide stronger evidence. That

is also the reason we collect multiple datasets within the five-year period over the source region (namely northern Africa), the Atlantic Ocean and the Amazon Basin. The model evaluation includes the PSMD scheme of dust upon emissions, the AOD over northern Africa and the Atlantic Ocean, the decline rate of AOD alone the trans-Atlantic transport as well as PM10 and coarse aerosol mass concentrations in the Amazon Basin (one site near the coast and one in the central basin) so that the constrain is conducted along the transport from source regions to receptor regions. In the revised manuscript, we also adopt **one more AERONET site over the east of Atlantic Ocean (Capo_Verde site in Fig. 3)** for the model evaluation.

In addition, we revise the manuscript to have more discussion regarding the dust simulation in the Amazon Basin in comparison with previous results. For example,

in line 333-342: "**Based on the regression line between observed concentrations of PM$_{10}$ and dust at the same site, Prospero et al. (2020) obtained a regional background value of PM$_{10}$ ranging from 17 to 22 μg m$^{-3}$, largely attributed to sea salt aerosol, and a value of 0.9 for the slope, suggesting PM$_{10}$ values above this range as a proxy for advected dust. Consistent with their results, the regression line between simulated dust and PM$_{10}$ from V12_C in this study shows a background value of PM$_{10}$ around 23 μg m$^{-3}$, with a value of the slope around 1.0, and the dust contribution to PM$_{10}$ is around 53% ± 20%. In contrast, the regression lines from V12 and V12_F are much steeper, with the slope of 1.4 and 2.1, respectively, and the dust contributions are relatively smaller, 44% in V12 and 34% in V12_F.**"

in line 474-484: "**Rizzolo et al. (2017) conducted aerosol measurements at ATTO from 19 March to 24 April 2015. The study showed the arrival of African dust between 3 and 6 April when the highest concentrations of PM$_{10}$, soluble Fe (III) and Fe (II) were recorded at ATTO. The peak value of 23 μg m$^{-3}$ for PM$_{10}$ was observed on 5 April. This dust event is well reproduced in this study with the peak value of 28 μg m$^{-3}$ for PM$_{10}$ on the same day and the dust contribution to PM$_{10}$ reaching above 70%. The co-occurrence of elevated sea salt concentration (reaching 2.5 μg m$^{-3}$) during this event is also found in this study, consistent with previous studies which show mixed transport of African dust and marine aerosol**

**to the basin (Wang et al., 2016; Ben-Ami et al., 2010; Rizzolo et al., 2017; Adachi et al., 2020).**"

Also, as suggested by the reviewer not to cover too many aspects, in the revised manuscript (Section 4.1 Dust emissions) we delete the discussion on the trend of annual dust emission and have more discussion on the impacts of Met. Field on dust emission fluxes. For example, in line 405-411: "**Fiedler et al. (2013) also found a maximum of emission flux over the Bodélé Depression in winter and the highest emission flux in spring in west Sahara. The study suggested that near-surface peak winds associated with Nocturnal Low-Level Jets is a driver of mineral dust emissions. Negative correlation between dust emissions and soil moisture has also been revealed by Yu et al. (2017) and Pierre et al. (2012), as the decreased vegetation growth in response to dry soil would result in enhanced dust emissions.**"

*Furthermore, the study heavily relies on the GEOS-Chem model results and only uses a few MODIS or AERONET observations for model evaluation. Especially, the AERONET sites and their available data are not good enough in spatial and temporal coverages, the major results or those statistical results/values are mainly calculated from model results (GEOS-Chem), which means most of these conclusions are model-dependent. It is important to note that changing or switching to another global transport model or using other dust schemes can significantly alter the results, and the study's major conclusions could be affected. Therefore, instead of focusing on the exact model values, the study should provide more accurate estimates about the relative values. (e.g. how much of the contribution from dust compare with other aerosols on the Amazon Basin; It there any interannual variability about the contribution, how significant it is? Which are the major factors impacting on the interannual variability, Met. conditions in transport or in emission flux). Also, better to highlight the points and conclusions from the study, not just describe the values and figures.*

Thanks for the nice comments. As suggested by reviewer, we have re-organized the structure of the manuscript and modified the corresponding discussion to highlight the

conclusions from the study. We now present more results about the dust contribution relative to other aerosols (including sea salt) and the interannual variability in section 4.3.1 and 4.3.2 in the revised manuscript. For example,

in line 478-484: "**This dust event is well reproduced in this study with the peak value of 28 μg m$^{-3}$ for PM$_{10}$ on the same day and the dust contribution to PM$_{10}$ reaching above 70%. The co-occurrence of elevated sea salt concentration (reaching 2.5 μg m$^{-3}$) during this event is also found in this study, consistent with previous studies which show mixed transport of African dust and marine aerosol to the basin (Wang et al., 2016; Ben-Ami et al., 2010; Rizzolo et al., 2017; Adachi et al., 2020)**"

in line 501-510: "**The dust contribution to surface aerosol concentrations averaged over the whole basin is 40% ± 4.5%, again with the maximum of 48% found in 2015. The location with the largest dust contributions (up to 70% in the north corner) slightly shifted inland compared to the spatial distribution of dust concentration. This could be explained by relatively higher influence of sea salt aerosols along the coast (around 30-50% near the coast of South America). Although the emission fluxes of both sea salt and dust are largely determined by surface winds, the interannual variability of dust concentrations is larger than sea salt over the Amazon Basin (20% vs. 10%) as the former is also sensitive to the export efficiency across the Atlantic Ocean as discussed above.**"

We also add the discussion on the dust contribution in Section 3 (model evaluation). For example, in line 337-342: "**Consistent with their results, the regression line between simulated dust and PM$_{10}$ from V12_C in this study shows a background value of PM$_{10}$ around 23 μg m$^{-3}$, with a value of the slope around 1.0, and the dust contribution to PM$_{10}$ is around 53% ± 20%. In contrast, the regression lines from V12 and V12_F are much steeper, with the slope of 1.4 and 2.1, respectively, and the dust contributions are relatively smaller, 44% in V12 and 34% in V12_F.**"

The discussion about the impact of Met. conditions and emission flux on dust concentrations reaching the Amazon Basin is presented in line 485-496 and line 441-455 in the revised manuscript. We also add more discussion on the impact of Met condition on the seasonality of dust lifetime. For example, in line 456-462: "**The**

**seasonality in the deposition fluxes and the consequent dust lifetime depends not only on precipitation but also the vertical pathways of dust transport across the Atlantic. Dust aerosols aloft at higher altitude reach further west and have relatively longer lifetime. Significant differences in dust vertical distributions along the transport pathways have been revealed from the CALIOP measurements, which show that more dust is transported above 2km in summer while the dust layer is the shallowest in winter (Liu et al., 2012).**"

*For the PMSD/PVSD, the paper mainly considers the coarse aerosols, see L186: what is the paper r definition of coarse aerosol here, diameter >1 um? If it is, I don't think it can derive the sea salt aerosols, and how much of the impact from sea salt during this long-range transport has not been discussed. Therefore, the paper should clarify the definition of coarse aerosols and address the impact of sea salt aerosols during long-range transport. I would suggest making substantially modifications before submitting it again based on following comments.*

Sorry for the confusion. We have defined the definition of coarse and fine aerosols in line 157-158: "**sulfate-nitrate-ammonium aerosols in fine mode (≤ 1 μm in diameter), sea salt in both fine and coarse (> 1 μm in diameter) ..**". For the comparison of PMSD, only the data dominated by dust (with little influence of sea salt) is used in the study. We have modified the corresponding text in line 222-231 to put it more clearly: "**In addition, to minimize the influence of aerosols other than dust, only data dominated by dust (simulated dust contribution to column-integrated aerosols mass concentrations > 95%) is used for the comparison of PMSD. There are a few sites not far from the coast and could be influenced by sea salt. With the above data screening, the sea salt contribution to total aerosol mass is less than 0.5%. For the comparison of AOD, the criterion is less stringent to have more data points available and uses data dominated by coarse aerosols (the contribution of fine aerosol to total aerosol volume < 3%). This criterion does not exclude sea salt and the contribution of sea salt to AOD could be up to 30% at the Capo_Verde site (22.9° W,16.7° N) over the east of the Atlantic Ocean.**" In addition, as replied to the previous comments, we have added more discussion on the influence of sea salt in the revised manuscript.

*Specific comments:*

1.  *The Abstract is not concise enough, somehow looks like introduction. I recommend revising the abstract to make it more concise and focused on summarizing the key points of the paper.*

    Thanks for the comments. We now revise the abstract accordingly.

2.  *Section 2.1, this study is using GEOS-Chem to simulate dust, the descriptions about dust scheme and the major factor controlling the emission (from the formula of dust emission flux) need to be discussed in the section.*

    Thanks for the comment. We have re-organized the structure of the manuscript and have new **Section 2.1.2 "dust emission and PMSD schemes in the model"** to have detailed descriptions of dust scheme in the model, including the major factor controlling the emissions in line 175-181: "**The emission of mineral dust is based on the dust entrainment and deposition (DEAD) mobilization scheme of Zender et al. (2003) in the GEOS-Chem model. The DEAD scheme calculates the total vertical dust flux based on the total horizontal saltation flux ($Q_s$) using the theory of White (1979). The $Q_s$ depends mainly on the surface wind friction velocity and the threshold friction velocity, which is determined by soil type, soil moisture content, and surface roughness. For more details of the DEAD scheme, readers are referred to Duncan Fairlie et al. (2007).**"

3.  *P6, L177: Why did the paper choose the 675 nm AOD from AERONET, not 550 nm, which is normally used for AOD comparison with observation?*

    The AOD from AERONET is available at wavelength of 440 nm and 675 nm, instead of 550 nm. Therefore, we choose the wavelength of 675 nm to perform the AOD comparison with the AERONET data. As suggested by the reviewer,

we choose AOD at 550 nm in the rest of the discussion (e.g. Section 4.3.2 AOD) in the revised manuscript.

4. *1: Which month?*

Figure 1 is for simulated annual dust emissions averaged over the year of 2013-2017. We now modify the caption of Figure 1 to clarify it: "**Figure 1.** Simulated **annual** dust emissions in GEOS-Chem, averaged from 2013 to 2017..."

5. *2: Is it surface wind or 10-m wind?*

It refers to 10-m wind. We now modify the caption of Figure 6 in the revised manuscript (Figure 2 in original version) to clarify it: "**Figure 6.** Monthly dust emission fluxes together with the 95$^{th}$ percentile hourly **10-m** wind speeds (red solid lines), the 75$^{th}$ percentile hourly **10-m** wind speeds (red dotted lines)…". We also modified the corresponding text to specify it.

6. *3 and L343: The V12 looks quite comparable as the observation for bin 2 and bin3, while V12_C is much better for bin1 and bin4, how did the paper conclude that the v12_C agree better with the observation?*

We now modify the corresponding text to quantify the comparison (line 279-281): "**The comparison indicates the model results based on V12_C agrees better with the observations, with much smaller mean absolute deviation (MAD) of 2.8, followed by 4.2 for V12 and 18 for V12_F.**"

7. *4: The quality of this figure is not clear enough, which is difficult to distinguish each experiment, the lines are almost overlapped by each other.*

We now re-draw the figure, in which the lines for observations are removed while the lines for model results are thicker.

8. *L261: the impact of Met. Fields on dust long-range transport need to be separated as two aspects: 1) the impact on dust emission flux, which is mainly related to the Met. Fields associated with the dust scheme used in GEOS-Chem (The standard dust scheme in GEOS-Chem is the dust entrainment and deposition (DEAD) mobilization scheme of Zender et al.[2003], combined with the source function used in the Global Ozone Chemistry Aerosol Radiation and Transport(GOCART) model [Ginoux et al., 2001;Chin et al., 2004]as described by Fairlie et al. [2007]). So the paper need to get the real Met. Fields in the emission formula to determine the correlations, I think here it is 10-m wind and soil moisture, please make it accurate and clearly. 2) The impact on dust transport (include deposition), especially long-range transport, including vertical velocity, precipitation, and wind.*

Thanks for the insightful comment. The DEAD and GOCART are two different schemes: the former is based on the theory of White (1979) and the latter is based on Gillette and Passi (1988) in computing the vertical dust flux. The met fields used in the calculation is from the assimilated met fields, namely GEOS-FP. As replied to the previous comments, we have re-organized the structure of the manuscript and have new **Section 2.1.2 "dust emission and PMSD schemes in the model"** to provide a detailed description of the dust emission scheme in the model, including the major factor controlling the emissions. When discussing the impact of Met. Fields on dust emission, we modify the text in line 394-396 to make it clearly that surface wind speeds used in the correlation analysis refers to 10-m wind: "Correlation analysis between dust emissions and meteorological variables suggests that the seasonality is mainly driven by high surface wind speeds (with *r* of 0.79-0.96 and 0.68-0.97 for the 75th and 95th percentiles of **10-m** wind speeds, respectively)."

We also add more discussion on the impact of Met. Fields on dust transport including precipitation and vertical pathways in line 456-462: "**The seasonality in the deposition fluxes and the consequent dust lifetime depends not only on precipitation but also the vertical pathways of dust transport across the Atlantic. Dust aerosols aloft at higher altitude reach further west and have relatively longer lifetime. Significant differences in dust vertical distributions along the transport pathways have been revealed from the CALIOP measurements, which show that more dust is transported above 2km in summer while the dust layer is the shallowest in winter (Liu et al., 2012)**"

9. *L272: what is the major Met. factor contributing to the significant emission decrease in 2013. The paper discussed the impact of the precipitation and other climate factors in the following descriptions by referring to some previous studies, but I would like to remind, for dust emission flux, only the 10-m wind and soil moisture are the major factors to impact on the emission flux (please double check the scheme formula), while the other Met. Fields or climate event are not directly impact on the emission flux in the emission formula, however, transport. If the paper would like to discuss the climate impact on those Met. Fields and transport, please use sensitivity experiments and provide more climate evidence from the model to validate it.*

The calculation of the dust emission flux is determined by wind friction velocity, soil moisture content, etc and are read from the assimilated Met. Fields (GEOS-FP). As replied to the previous comments, we have added **Section 2.1.2 "Dust emission and PMSD schemes in the model"** to make it more clearly. In addition, we now delete the discussion on the trend of annual dust emission for the following reasons: 1) the five-year period may be not long enough to derive the trend; 2) as suggested by the reviewer, we try not to

cover too many aspects and have more discussion on the interesting points (e.g. the impact of Met. Fields on the long-range transport of dust).

10. *L359 and Fig. 5: The total AOD includes all the aerosol species, how can the paper get accurate estimate about dust AOD biases and its PMSD? I do agree that the modeled AOD is much lower than that of the MODIS observation, especially over the downwind areas, dust may be one of the reasons, but it cannot conclude how much of the contribution is coming dust. Also, V12 shows low biases over the dust source region, while the V12_C shows much larger AOD over the dust source region of western Africa, but it also underestimates the AOD in the downwind areas between 40-60W, why? I am not sure why did the paper use log scale here for figure e and f, can the paper explain that, zoom in the differences? It is difficult for me to quantify the exact values between these different PMSD on AOD.  How important is the small AOD differences (less than 0.02-0.03) over the ocean?*

Sorry for the confusion. As replied to the previous comments, PMSD comparison is conducted only for data dominated by dust and thus has little influence of other aerosols. We also use mean absolute deviation to quantify the comparison between observations and different model schemes.

For AOD comparison with MODIS AOD, we have added the following discussion to explain the comparison in line 305-310: "**Note that the model results based on V12_C tends to overestimate MODIS AOD over Africa while no significant systematic bias is found between V12_C and AERONET AOD. Wang et al. (2016) sampled MODIS data at AERONET sites over Africa and found that MODIS retrieval underestimate AERONET AOD at most sites with NMB of -12% − -36%, which partly explain the large difference between model V12_C and MODIS AOD.**"

Figure 4 in the revised manuscript (Fig. 5 in original draft) is used for the comparison of the removal rates of aerosol along the trans-Atlantic transport,

which could be represented by the slope of the regression line based on log(AOD) instead of AOD. We have modified the corresponding text to make it more clearly in line 311-319: "**Assuming first-order removal of aerosol along the transport, we could derive the removal rates of aerosols, estimated as the gradient of the logarithm of AOD (log(AOD)) against the distance over the Atlantic Ocean along the transport path (AOaTP, 20° − 50° W and 5° S − 25° N, Figure 4e). The decline rate of MODIS log(AOD) is 0.019 ± 0.0025 degree$^{-1}$. A similar decline rate of 0.019 ± 0.0029 degree$^{-1}$ is found for simulated log(AOD) based on V12_C. In contrast, simulations with V12 and V12_F exhibit relatively steeper slopes of 0.021 ± 0.0040 and 0.021 ± 0.0041, respectively, implying too much aerosol removal and thus lower export efficiency along the transport.**"

11. 6: Again, better to derive the contributions from other aerosols when the paper compares the dust concentration with the observed PM10.

    As replied to the previous major comments, we modified the corresponding text to to include the results about the dust contribution.

12. *Section 4, which PMSD scheme did the paper use in the analysis of this section?*

    The results are based on V12_C. We have re-organized the structure of the manuscript and moved the model evaluation to **Section 3 (Model evaluation)**, at the end of which we put it clearly that "**we use the model results from V12_C (hereinafter referred to as model results unless noted otherwise)** to investigate the transatlantic transport of dust from Africa and its impact over the Amazon Basin in the following sections."

13. *Table S2, this is dry deposition or wet deposition?*

We modify the caption of the table to make it clearly: "Seasonal dust deposition (**including dry and wet deposition**, Pg yr-1)"

14. *Figure S4: I don't think one seasonal average figure of column burden can clearly describe the accurate transport path. If the paper would like to discuss the transport path, please use a more accurate analysis of aerosol horizontal/vertical fluxes or divergence analysis, cross-section analysis with temporal evolution. Also, the transport path at different layers would be quite different (a lot of previous studies have shown that), please don't get the conclusions without providing enough support. If the paper cannot provide evidence to support the conclusions, I would suggest not to include them in the paper. The paper has included many of these descriptions/conclusions without providing strong support from both model and observation analysis.*

Thanks for the nice suggestion. We now delete this figure and modify the corresponding discussion in the revised manuscript: "**The amount of African dust reaching the Amazon Basin depends not only on the dust emission fluxes, but also the transport path. Associated with the annual oscillation of ITCZ, the outflow of African dust moves slightly southwest toward South America in boreal winter and spring, and moves west towards the Caribbean in boreal summer and fall (Moran-Zuloaga et al., 2018; Ben-Ami et al., 2012). Therefore, although higher dust load over the coastal region of North Africa is found in summer (> 500 mg m$^{-2}$), dust reaching the Amazon Basin is less than 10 mg m$^{-2}$. In contrast, dust load over the Amazon Basin could reach up to 50 mg m$^{-2}$ in spring and winter…**"

15. L428: Better to show the formula about the way to calculate the life time.

we now add the equation in the corresponding text: "Assuming first-order removal of dust aerosols, we further derived seasonal e-folding lifetime (hereinafter referred to as lifetime, $\tau$) of simulated dust during $2013 - 2017$,

based on the logarithm of the dust column burden against travel time over the AOaTP (Figure 7) **using Equation 1:**

$$\tau = \frac{L}{v \times slope} \qquad\qquad (1)$$

**where L is the distance of 1-degree longitude averaged over 5° S − 25° N in unit of m degree-1; $v$ is the wind speed in unit of m s-1; and slope is the gradient of the linear trend line based on the logarithm of dust burden against the distance in degree between 20 ºW and 50 ºW."**

16. *L438: How did the paper define/calculate the dust deposition flux and wet deposition ratio? I saw that the wet deposition ratio is not the largest in winter, which is different to the descriptions in L435.*

    We now modify the text to specify that deposition includes both dry and wet deposition: "The short lifetime in winter is generally associated with high deposition flux (**including both dry and wet deposition**)…". We also change the subtitle in Figure 8 from "wet deposition ratio" to "**contribution of wet deposition**". The lifetime is determined by the total deposition, therefore, the relatively high deposition flux in winter is consistent with the shorter lifetime in winter.

17. *Section 5: which PMSD scheme did the paper use in the analysis of this section?*

    Sorry for the confusion. The results are based on V12_C. As replied to previous comments, we have re-organized the structure of the manuscript and put it clearly that "**we use the model results from V12_C (hereinafter referred to as model results unless noted otherwise)** to investigate the transatlantic transport of dust from Africa and its impact over the Amazon Basin in the following sections."

18. *Figure 9: I am confused about this figure, better to describe it with more details. For dust and sea salt, is it total concentration or coarse part? Also, what is the major points about this figure, I saw a lot of descriptions in the section with numbers/values based on the model results, I am wondering how much we can trust them and how did the paper use the observation to validate its performance since I did saw many biases between the observation and model in this figure? What is "Other" meaning in the figure? I can get the information about the "good performance" from the figure showing here.*

Sorry for the confusion. The dust and sea salt plotted in the figure is total concentration. We modified the text in line 156-160 in Section 2.1.1 (Model overview) to provided more detailed description of aerosols simulated in the model: "**The aerosol simulation is an offline simulation for aerosol tracers including black carbon (BC), organic aerosols (OA), and sulfate-nitrate-ammonium aerosols in fine mode ($\leq$ 1 μm in diameter), sea salt in both fine and coarse (> 1 μm in diameter) modes, and mineral dust in four size bins covering the size range of 0.2 – 12 μm in diameter.**" We also modified the legend in the figure to use "**sulfate-nitrate-ammonium**" instead of "Other" and "**OA**" instead of "OC".

The main purpose of the figure here is to show the time series of surface aerosol mass concentrations and specifically the frequent peaks driven by dust intrusion. As replied to the previous comments, we modified the corresponding text to make it concise and to the point.

19. Section 5.2 and Fig.10: can the paper describe how to calculate the dust events frequency and the interannual variation of dust events? Also, I really don't know what is the point about analyzing them? Any interesting points that the paper would like to highlight here? This is dust intrusion pattern due to long-range transport. Also, for figure (c) and (d), which season, please clarify these details clearly in several places?

For the calculation of the dust event frequency, we modified the text in line 511-514 to make it clear: "**Figure 10c also shows the frequency of dust events over the Amazon Basin, estimated as the number of days when daily surface dust concentrations reaching the threshold of 9 μg m$^{-3}$ (Moran-Zuloaga et al., 2018) divided by the total number of days in the wet season of 2013 – 2017.**"

The interannual variation of dust events is estimated as relative standard deviation (RSD). We also modify the text in line 517 to make it clear: "The interannual variation of the frequency (**represented by RSD**), however,…"

We now modify the figure caption to clarify these details and also present dust contribution to surface aerosol concentrations during dust event instead of the interannual variation of dust events for figure (d). We also modify the corresponding text to discuss the dust contribution to surface aerosol concentrations during dust events in comparison with the condition over the whole wet season: "**Dust frequency averaged over the whole region is around 18% ± 4.6% and decreases from 50 − 60% at the northeast coast to < 1% in southern inland. The frequency of dust events at ATTO site is around 32%, close to the median of the range. The interannual variation of the frequency (represented by RSD), however, has an opposite trend, gradually increasing from 10% at the northeast coast to over 100% in southern inland (36% at ATTO). During dust events, the dust mass concentration at ATTO reaches 16 ± 2.9 μg m$^{-3}$ (three times as high as that over the whole wet season), accounting for around 75% ± 5.3% of total aerosol (Figure 10d)…**"

20. Section 5.3 and Figure 12: I agree this is a useful and interesting application to estimate the nutrient input based on previous studies or measurements. But I would like to emphasize, the accuracy of these conclusions should be based on how much we can trust the model results. In the other words, the model

performance of dust needs to be validated as pretty good performance from different aspects, but I don't think this part has been done well in the paper to provide enough evaluations.

We agree with the reviewer that the conclusion is based on how much we can trust the model results. As replied to previous comments, we have modified the manuscript according to the reviewer's comments to demonstrate the model performance of dust simulation regarding its PMSD, trans-Atlantic transport efficiency, and concentrations over the Amazon basin constrained with multiple observation datasets (summarized in Table 2). We also have more discussion on the comparison with previous results to show the consistence between our results and previous observations, e.g. in line 474-484: "**Rizzolo et al. (2017) conducted aerosol measurements at ATTO from 19 March to 24 April 2015. The study showed the arrival of African dust between 3 and 6 April when the highest concentrations of $PM_{10}$, soluble Fe (III) and Fe (II) were recorded at ATTO. The peak value of 23 $\mu g\ m^{-3}$ for $PM_{10}$ was observed on 5 April. This dust event is well reproduced in this study with the peak value of 28 $\mu g\ m^{-3}$ for $PM_{10}$ on the same day and the dust contribution to $PM_{10}$ reaching above 70%. The co-occurrence of elevated sea salt concentration (reaching 2.5 $\mu g\ m^{-3}$) during this event is also found in this study, consistent with previous studies which show mixed transport of African dust and marine aerosol to the basin (Wang et al., 2016; Ben-Ami et al., 2010; Rizzolo et al., 2017; Adachi et al., 2020).**"

For the nutrient input, we also summarize the results from previous studies (Table 4) and discuss the possible factors contributing to the uncertainties in those results in line 570-588 and also in line 601-605. We also add the following sentence in line 605-608 to emphasize the necessity of observational constraints on model results: "**More observations including the mass fraction of nutrient in dust aerosols and the deposition fluxes of those**

**elements in the Amazon Basin are necessarily required in the future work to better evaluate the nutrient input associated with the African dust intrusion.''**

---

## Referee Report (RR1)

Review of: The export of African mineral dust across the Atlantic and its impact over the Amazon Basin

By:

Wang et al.

General Comments:

The authors have provided a thoughtful consideration of my earlier comments. I am happy with the overall quality of the manuscript. I still recommend a few adjustments in the minor comments, but I will be happy to recommend publication after these small suggestions are at least considered.

Specific comments:

l. 541  It made me confused when I read a DOD value higher than the AOD. I went to the references mentioned and saw that the DOD of 0.18 was found in the strongest dust event measured, while the AOD of 0.14 is the average value during dust events. Maybe you should rewrite this sentence to make this clearer, as now it might be confusing.

l. 560  I think you meant "exists" instead of "exits".

l. 642  You discuss the matter of the compensation of hydrologic losses of nutrients by Saharan dust in section 4.3.3. In the paragraph starting in line 589 you present values and discuss uncertainties. You discuss that the dust nutrient input could compensate for the hydrologic losses, but the amount of compensation depends on a few uncertainties. Just by comparing the estimated hydrologic losses presented in that paragraph with the estimated inputs you show in line 641, one sees that the dust nutrient inputs might not be enough to compensate for the losses all by itself. As you correctly wrote in line 607, more work is required in order to have clearer answers on that. For that reason, I would be more careful with the wording in line 642. "Which may well compensate" might lead the reader to conclude that the dust nutrients input is more than enough, while your own results show that this might not be the case.

Table 3.        I would include "estimated using (...)" in the title. To clarify that these are results of simulations under certain conditions. This information is already in the text, but I believe it is important to have this kind of information in the title or legend of tables or figures.

---

## Referee Report (RR2)

I appreciate the authors' effort in revising the manuscript, particularly in response to the reviewers' comments, which necessitated a significant restructuring and reorganization of the paper. Most of my remaining comments have been satisfactorily addressed. However, the paper could benefit from a thorough grammar review. Nonetheless, I still have some minor comments below, and once these are addressed, I find the paper may be suitable for publication.

1. Abstract: Consider using relative percentage values instead of concentration values for improved clarity.
2. Introduction: L110, I don't think it is necessary to mentioned other species of biomass burning tracers and sea salt here.
3. Introduction, L135: please add the descriptions about the contents in each section.
4. L157: Delete the word "and" to enhance sentence flow.
5. Section 4.2, L428: It is advisable to emphasize that these values are GEOS-Chem model-based and subject to model dependency.

---

## Author Response (AR2)

**We thank the reviewers for their supportive and thoughtful comments. Our responses to the comments are provided below in a red font, with the reviewers' comments in a black italicized font.**

**Review #1**

*General Comments:*
*The authors have provided a thoughtful consideration of my earlier comments. I am happy with the overall quality of the manuscript. I still recommend a few adjustments in the minor comments, but I will be happy to recommend publication after these small suggestions are at least considered.*

Thanks for the reviewer's support. We have addressed all the concerns. Please see our point-by-point response below.

*Specific comments:*
*l. 541 It made me confused when I read a DOD value higher than the AOD. I went to the references mentioned and saw that the DOD of 0.18 was found in the strongest dust event measured, while the AOD of 0.14 is the average value during dust events. Maybe you should rewrite this sentence to make this clearer, as now it might be confusing.*

Sorry for the confusion. We have revised this sentence into: "Consistent with our results, previous studies by Baars et al. (2011) and Baars et al. (2012) **reported an average AOD (532 nm) of ~0.14 when affected by the influence of Saharan dust at a similar Amazon site (60° 2.3′ W, 2° 35.9′ S), during which the DOD (532 nm) could be up to 0.18**."

*l. 560 I think you meant "exists" instead of "exits".*

Thanks for pointing out this mistake. We have corrected the typo.

*l. 642 You discuss the matter of the compensation of hydrologic losses of nutrients by Saharan dust in section 4.3.3. In the paragraph starting in line 589 you present values and discuss uncertainties. You discuss that the dust nutrient input could compensate for the hydrologic losses, but the amount of compensation depends on a few uncertainties. Just by comparing the estimated hydrologic losses presented in that paragraph with the estimated inputs you show in line 641, one sees that the dust nutrient inputs might not be enough to compensate for the losses all by itself. As you correctly wrote in line 607, more work is required in order to have clearer answers on that. For that reason, I would*

*be more careful with the wording in line 642. "Which may well compensate" might lead the reader to conclude that the dust nutrients input is more than enough, while your own results show that this might not be the case.*

Thank you for the nice suggestion. The estimated input of phosphorus is within the range of its *hydrologic losses but the input of* magnesium is much less than its *hydrologic losses*. Considering the uncertainties within the estimates, we agree with the reviewer that the wording *"Which may well compensate" might cause some misleading.* We have now modified the corresponding text in line 595-602 to make the description more clear: "On the other hand, Vitousek and Sanford (1986) reported a loss of $0.8 - 4$ mg m$^{-2}$ yr$^{-1}$ for phosphorus and 810 mg m$^{-2}$ yr$^{-1}$ for magnesium in Brazilian ecosystem to surface waters. **Estimated annual phosphorous deposition flux of $0.97 \pm 0.16$ mg m$^{-2}$ yr$^{-1}$ into the Amazon Basin on average in our study is at the bottom end of the range of its hydrologic losses, implying that the nutrient input from African dust could to a large extent compensate the hydrologic losses of phosphorous in Brazilian forest ecosystem, although the deposition input of magnesium is much less than its hydrologic losses.**"
We also revised the corresponding sentence (in both Abstract and Conclusion) from "which may well compensate" to "which may **to some extent** compensate"

*Table 3. I would include "estimated using (...)" in the title. To clarify that these are results of simulations under certain conditions. This information is already in the text, but I believe it is important to have this kind of information in the title or legend of tables or figures.*

Thanks for the comment. We have accordingly modified the title.

**Review #2**

*I appreciate the authors' effort in revising the manuscript, particularly in response to the reviewers' comments, which necessitated a significant restructuring and reorganization of the paper. Most of my remaining comments have been satisfactorily addressed. However, the paper could benefit from a thorough grammar review. Nonetheless, I still have some minor comments below, and once these are addressed, I find the paper may be suitable for publication.*

Thanks for the reviewer's thoughtful comments to help us improve the manuscript.

We have now addressed all the concerns, including the thorough grammar review.

Please see our point-by-point reply below.

1. *Abstract: Consider using relative percentage values instead of concentration values*

*for improved clarity.*

Thanks for the nice suggestion. We have revised the Abstract accordingly: "**African dust entering the Amazon Basin during the wet season accounts for 40% ± 4.5% (up to 70%) of surface aerosol mass concentrations over the basin**."

*2. Introduction: L110, I don't think it is necessary to mentioned other species of biomass burning tracers and sea salt here.*

Thanks for the comment. We have deleted the sentence.

*3. Introduction, L135: please add the descriptions about the contents in each section.*

Thanks for the comment. We have added the description accordingly at the end of this paragraph: "**The paper is organized as follows: Section 2 describes the model setup for dust simulation and the observational datasets applied to constrain the model results; Section 3 gives the model evaluation regarding the simulation of the export of African dust towards the Amazon Basin; Section 4 presents the model results, including simulated dust emissions in Africa, the trans-Atlantic transport of African dust, and the influence of African dust over the Amazon Basin; and Section 5 summarizes the main conclusions drawn from this study.**"

*4. L157: Delete the word "and" to enhance sentence flow.*

Done!

*5. Section 4.2, L428: It is advisable to emphasize that these values are GEOS-Chem model-based and subject to model dependency.*

Thanks for the nice comments. We revised the corresponding text into: "**The GEOS-Chem results in this study are consistent with this seasonal oscillation:** although higher dust load over the coastal region of North Africa is found in boreal summer (> 500 mg m$^{-2}$), dust reaching the Amazon Basin is less than 10 mg m$^{-2}$; In contrast, dust load over the Amazon Basin could reach up to 50 mg m$^{-2}$ in boreal spring and winter."